# *StatsMerging*: Statistics-Guided Model Merging via Task-Specific Teacher Distillation

## Abstract

As large models are increasingly deployed across various tasks, the limited GPU memory available for storing and executing task-specific models presents a growing bottleneck. Model merging has emerged as a promising solution to accommodate multiple large models within constrained memory budgets. While traditional multi-task learning methods attempt to merge shared layers, they require labor-intensive annotated labels and incur significant computational overhead. Recent merging techniques aim to address this issue by combining models at inference time; however, these approaches often rely on simplistic heuristics, ignore weight distribution characteristics, assume architectural identity, or require access to test samples to infer merging coefficients, thereby limiting their generalization capability and scalability. We present *StatsMerging*, a novel lightweight learning-based model merging method guided by weight distribution statistics without requiring ground truth labels or test samples. *StatsMerging* offers three key advantages: (1) It uniquely leverages singular values from singular value decomposition (SVD) to capture task-specific weight distributions, serving as a proxy for task importance to guide task coefficient learning; (2) It employs a lightweight learner *StatsMergeLearner* to model the weight distributions of task-specific pre-trained models, improving generalization and enhancing adaptation to unseen samples; (3) It introduces *Task-Specific Teacher Distillation* for merging vision models with heterogeneous architectures, a merging training paradigm that avoids costly ground-truth labels by task-specific teacher distillation. Notably, we present two types of knowledge distillation, (a) distilling knowledge from task-specific models to train *StatsMergeLearner*; and (b) for the first time, distilling knowledge from models with different architectures prior to merging, following a distill-then-merge paradigm. Extensive experiments across eight tasks demonstrate the effectiveness of *StatsMerging*. Our results show that *StatsMerging* outperforms state-of-the-art techniques in terms of overall accuracy, generalization to unseen tasks, and robustness to image quality variations.

## 1 Introduction

Computer vision has witnessed transformative progress fueled by deep learning, particularly through the development and adoption of large-scale pre-trained models. Architectures like Convolutional Neural Networks (CNNs) (Krizhevsky et al., 2012; He et al., 2016; Simonyan and Zisserman, 2014), Vision Transformers (ViTs) (Dosovitskiy et al., 2021b; Touvron et al., 2021), and hybrid approaches (Liu et al., 2022) pre-trained on massive datasets have become the cornerstone of modern vision applications. Large-scale models leveraging multi-modal pre-training, such as CLIP (Radford et al., 2021)) or generative models like GANs (Goodfellow et al., 2014) and Diffusion Models (Ho et al., 2020; Rombach et al., 2022) have further pushed the boundaries of visual understanding and synthesis,

Submitted to 39th Conference on Neural Information Processing Systems (NeurIPS 2025). Do not distribute.

enabling the use of pre-trained backbones to a wide range of downstream vision applications. The dominant practice is to fine-tune these powerful base models to computer vision tasks, including image classification (He et al., 2016), object detection (Ren et al., 2015; Carion et al., 2020a), semantic segmentation (Long et al., 2015; Xie et al., 2021), image restoration (Zhang et al., 2017; Saharia et al., 2022), and image generation (Mirza and Osindero, 2014). This success, however, leads to a practical challenge: the proliferation of numerous specialized pre-trained weights and model checkpoints (Cao et al., 2024a, 2025), most of which share the same foundational ViT or CNN backbones. Managing this growing collection incurs significant storage overhead, complicates deployment, and represents a missed opportunity to consolidate the related, yet specialized, knowledge contained within these models (Wortsman et al., 2022), particularly on compute-constrained platforms such as edge devices (Cao et al., 2024b; Singh et al., 2024). While Multi-Task Learning (MTL) (Vandenhende et al., 2022b) aims to create versatile single models for vision tasks, it often demands complex joint training strategies, concurrent access to diverse datasets, and careful architecture design to balance performance across disparate tasks.

Model merging offers a compelling post-hoc alternative, seeking to combine independently trained models without expensive retraining. However, while techniques for model merging have gained traction, particularly in Natural Language Processing (NLP) (Yadav et al., 2023a; Ilharco et al., 2023), adapting these techniques in computer vision domain has far less explored. A straightforward approach of simple weight averaging (Wortsman et al., 2022) often fails in vision tasks due to the complex, hierarchical visual feature representations, task-specific optimizations, and the presence of intricate noise patterns that lead to sharp, non-convex loss minima (Izmailov et al., 2018). Recent methods in this direction (Matena and Raffel, 2022; Jin et al., 2023; Yang et al.; Padmanabhan et al., 2023) neglect the importance of weight distribution.

This paper introduces a novel model merging framework specifically designed to address the afore-mentioned challenges within computer vision. We propose *StatsMerging*, a weight distribution statistics-guided merging approach that moves beyond simple parameter averaging or task-vector manipulation. *StatsMerging* leverages the statistical features models pre-trained on prior tasks for merged. In particular, we compute salient statistics extracted by leverage Singular Value Decomposition (SVD) to capture the dominant properties of the learned feature spaces. This statistical information, intrinsically capturing aspects of the pre-trained model distributions, guides the merging process by learning a compact Multilayer Perceptron (MLP), coined *StatsMergeLearner* that predicts adaptive merging coefficients ($\lambda$) shown in Fig. 1. This allows the merging to be guided by the weight landscape, rather than treating coefficients as free parameters requiring external tuning data.

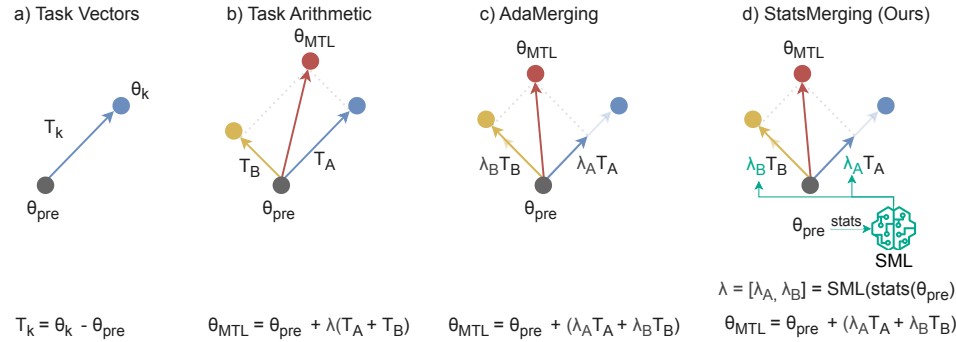

Figure 1: Compared to prior works, *StatsMerging* uniquely learns the merging coefficients using *StatsMergeLearner*, taking advantage of statistical features of weigts pre-trained on prior tasks. Notably, while both AdaMerging and *StatsMerging* are presented in the task-wise level in c) and d) for simplicity of illustration, the same principle can be applied at the layer-wise level for fine-grained adaptation.

We make four significant contributions summarized as follows:

- We propose *StatsMerging*[1], a novel model merging framework guided by model weight statistics, leveraging singular values extracted via Singular Value Decomposition (SVD) to predict merging coefficients $\lambda$.

---

[1]Our code is available at https://github.com/statsmerging/statsmerging.

- We design the lightweight *StatsMergeLearner* to learn model merging coefficients $\lambda$ estimation based on weight statistical features, through a newly proposed Task-Specific Teacher Distillation training paradigm without manually-annotated labels.

- We introduce the first heterogeneous architectural merging method, which distills knowledge from models with non-identical architectures into the unified target architecture.

- Extensive experiments demonstrate the effective of our proposed *StatsMerging*, achieving 84.5% average accuracy on merging models from eight tasks, outperform the state-of-the-art AdaMerging (81.1%) by a substantial margin of 3.4%.

## 2 Related Work

### 2.1 Multi-Task Learning

Multi-Task Learning (MTL) (Zhang and Yang, 2021; Vandenhende et al., 2022a) represents a paradigm for training a single model to perform multiple tasks concurrently. While MTL aims to create unified models capable of handling diverse objectives, it typically requires careful design of network architectures, computationally expensive training, access to large and diverse datasets, and intricate task balancing strategies (Zhang and Yang, 2021). Furthermore, MTL necessitates joint training from the outset, which can be computationally expensive and may not be feasible when dealing with a collection of pre-trained, specialized models. Model merging offers a compelling alternative by enabling the combination of independently trained models, without the need for extensive retraining or simultaneous access to multi-task datasets. Our work distinguishes from MTL by focusing on efficiently transferring existing knowledge within specialized models through weight statistics-guided merging rather than joint training.

### 2.2 Multi-Task Merging

Early approaches to model merging often involved simple heuristics like Weight Averaging (Wortsman et al., 2022), Ties-Merging (Yadav et al., 2023a), and Arithmetic Merging (Ilharco et al., 2023). While straightforward to implement, these methods (Ye et al., 2023; Akiba et al., 2025; Tang et al., 2025) typically lack awareness of the weight distributions and learned representations within the models, leading to suboptimal performance in the merged model compared to individually fine-tuned models or unified models trained from scratch. For instance, Wortsman et al. (Wortsman et al., 2022) demonstrated that naive weight averaging could significantly degrade performance, highlighting the challenges in consolidating knowledge from independently trained networks. More recent methods, such as those explored in natural language processing (Yadav et al., 2023b; Ilharco et al., 2023), have shown promise by learning interpolation weights. However, these often treat the weights as free parameters, potentially requiring significant tuning data and not explicitly leveraging the weight distribution of the models being merged, a key distinction from our proposed approach. The gap often lies in effectively unifying the diverse and task-specific feature representations learned by individual models into a single, high-performing entity without extensive retraining.

### 2.3 Merging Methods in Computer Vision

The application of model merging techniques in computer vision is relatively less explored compared to natural language processing (Yadav et al., 2023b; Ilharco et al., 2023). Computer vision models, particularly deep convolutional neural networks (CNNs) (Krizhevsky et al., 2012; He et al., 2016; Simonyan and Zisserman, 2014) and Vision Transformers (ViTs) (Dosovitskiy et al., 2021a; Touvron et al., 2021), learn complex, hierarchical feature representations that are highly sensitive to task-specific optimizations (Izmailov et al., 2018). Simple averaging techniques often fail due to the non-convex nature of the loss landscape and the divergence of learned feature spaces across different visual tasks. Recent advancements (Matena and Raffel, 2022; Yang et al.) have shown potential, but often lack explicit mechanisms to account for the unique properties inherent in visual data and architectures, such as spatial relationships in CNNs or attention mechanisms in ViTs. Furthermore, the effectiveness of these methods across the broad spectrum of computer vision tasks, including low-level restoration (Zhang et al., 2017; Saharia et al., 2022), mid-level detection (Ren et al., 2015; Carion et al., 2020b), and high-level classification (He et al., 2016), has not been comprehensively validated. Our work addresses these limitations by introducing a novel merging framework that

leverages internal model weight statistics to guide the merging process, making it more adaptable and effective across diverse computer vision tasks and architectures.

| Method | No Manual Label | No TT Samples | Layer Level | TT Adaptability | Heterogeneous Architecture |
|---|---|---|---|---|---|
| Traditional MTL | ✗ | ✗ | * | ✗ | ✗ |
| Task Arithmetic | ✓ | ✓ | ✗ | ✗ | ✗ |
| Ties-Merging | ✓ | ✓ | ✗ | ✓ | ✗ |
| Fisher Merging | ✓ | ✓ | ✗ | ✗ | ✗ |
| RegMean | ✓ | ✓ | ✗ | ✗ | ✗ |
| AdaMerging | ✓ | ✗ | ✓ | ✓ | ✗ |
| *StatsMerging* (Ours) | ✓ | ✓ | ✓ | ✓ | ✓ |

Table 1: Summary of system characteristics in recent works. *: Optional. TT: Test-Time.

In summary, our method *StatsMerging* enjoys several advantages compared to prior works: (1) no human annotated labels are required to construct the training set; (2) no validation samples are needed to compute the weight coefficients for merging; (3) it works in the Layer-Wise level; (4) it allows for test-time adaptability; (5) it can extend to heterogeneous architectures.

## 3 Methodology

### 3.1 Preliminaries

**Notations:** A deep neural network is parameterized by a set of weights $\theta = \{\theta_1, \theta_2, \ldots, \theta_L\}$ that learns the mapping from an input data $x_i \in \mathbb{R}^d$ to a predicted value $\hat{y}_i \in \mathbb{R}^D$: $f_\theta(x_i) \to \hat{y}_i$. Of these, $\theta^l$ represents the $l$-th $l \in \{1, 2, \ldots, L\}$ layer weights where $L$ is the number of layers of the model $f_\theta$, $d$ denotes an input data $x_i$'s dimension. For classification problems, $y_i$ is the class label and $D$ is the number of classes, while for regression problems, $D$ is the dimension of the output vector $y_i$.

The weights of a pre-trained model (e.g., ViT or ResNet) are denoted as $\theta_{pre} = \{\theta_{pre}^1, \theta_{pre}^2, \ldots, \theta_{pre}^L\}$. The weights fine-tuned on a specific training data $\{x_i, y_i\}_{i=1}^{N_k^{tr}}$ for task $k$ is recorded as $\theta_k = \{\theta_k^1, \theta_k^2, \ldots, \theta_k^L\}$ where $N_k^{tr}$ is the number of training samples.

**Problem Formulation:** The problem of *model merging* is formulated as given $K$ tasks' training data, find a way to combine weights $\{\theta_k\}_{k=1}^K$ fine-tuned for $K$ tasks previously to obtain a new weight $\theta_m$ without undergoing the retraining process, while the new model $f_{\theta_m}$ is capable of performing well on $K$ tasks jointly.

It is assumed that all $K$ fine-tuned weights and the merged weight share the same neural network architecture. Therefore, the core question is how to *linearly combine* $\{\theta_k\}_{k=1}^K$ to obtain $\theta_m$. In the task level, the model merging problem is finding a set of coefficients $\lambda_k \in \{\lambda_1, \lambda_2, \ldots, \lambda_K\}$ such that the merged model weights $\theta_m = \sum_{k=1}^K \lambda_k \theta_k$ for model $f_{\theta_m}$ perform well on all $K$ tasks. In the layer level, it becomes searching for a set of coefficients $\lambda_k^l \in \{\lambda_1^1, \lambda_1^2, \ldots, \lambda_1^L, \lambda_2^1, \lambda_2^2, \ldots, \lambda_2^L, \ldots, \lambda_K^L\}$ to obtain the merged model $\theta_m = \sum_{k=1}^K \sum_{l=1}^L \lambda_k^l \theta_k^l$ that maintain high performance on $K$ tasks.

### 3.2 Weight Statistics-Guided Model Merging

In this section, we describe the main intuition and techniques of our proposed method: *StatsMerging*. Our core idea is that given the distribution of pre-trained weights $\theta_k$, we can learn a function $g(\theta_k) \to \lambda_m$ to predict the merging coefficients $\lambda_m$. We argue that *weight distribution* plays an important role in model merging. However, directly using the raw weights $\theta_k$ as input is impractical due to the high dimension of $\theta_k$. We posit that such information can be represented by weight statistics. These statistical features contain key information regarding the amount of weights $\theta_k$ for a task $k$ to be merged to the final model. We highlight the key differences with prior works in Fig. 1 d).

**Weight Statistics:** For a pre-trained weight $\theta_k$ on task $k$, we compute the mean $\mu_{\theta_k}$ and variance $\sigma^2 = Var(\theta_k)$ to represent its center and breadth, as well as its magnitude $m = ||\theta_k||$. In addition,

we extract the singular values $\sigma_i'$ from Singular Value Decomposition (SVD):

$$W_k = U_k \Sigma_k V_k^\top \tag{1}$$

where $W_{\theta_k}$ represents the matrix of the model parameter $\theta_k$. By default, we use rank 3 from $\Sigma_k$ to form weight statistics. We hypothesize that singular values compress the key information regarding weight distribution that can benefit the decision of assigning the amount of weights from $\theta_k$ for merging. Combining all together, the weight statistics feature vector $S_k$ is formed as

$$S_k = stats(\theta_k) = [\mu, \sigma^2, m, \sigma_r'] \tag{2}$$

where $stats()$ extracts the statistical features from the weight $\theta_k$, $\sigma_r$ represents the singular value vector given rank $r$: $\sigma_r' = [\sigma_1', \sigma_2', \ldots, \sigma_r']$.

Notably, the Equation 3 above is task-wise while we also introduce layer-wise formulation for layer $l$:

$$S_k^l = stats(\theta_k^l) = [\mu, \sigma^2, m, \sigma_r']^l \tag{3}$$

where the layer-wise statistics features of pre-trained model from task $k$ layer $l$ is computed.

**StatsMergeLearner (SML):** We adopt a multilayer perceptron (MLPs) to learn to predict the merging coefficients $\lambda$ given weight statistics feature vector $S_k$ as input. In the task-wise mode, the *StatsMergeLearner* is denoted as $SML(S_k)$:

$$\lambda_k = SML(S_k) = g(stats(\theta_k)) \tag{4}$$

where $\lambda_k$ is a scalar representing the merging coefficient of Task $k$ model. In the layer-wise mode, the *StatsMergeLearner* is denoted as $M(S_k)$:

$$\lambda_k^l = SML(S_k^l) = g(stats(\theta_k^l)) \tag{5}$$

where $\lambda_k$ is a vector containing $L$ layers' coefficients and $\lambda_k^l$ refers to the coefficient of layer $l$ in the $k$ pre-trained model. By default, we use a two-layer MLP to implement the *StatsMergeLearner*.

**Optimization Objective.** To train *StatsMergeLearner*, in the standard supervised training paradigm, we freeze the weights for each task $\theta_k$ and apply the cross-entropy loss function $L_{CE}$ on the aggregated dataset:

$$\mathcal{L}_{\text{CE}}^{SL} = -\sum_{c=1}^{C_m} y_c \log(\hat{y}_c)) \tag{6}$$

where $\hat{y}_c$ is the prediction from the merged model for class $c$, $C_m$ is the total number of classes in the aggregated dataset.

### 3.3 Task-Specific Teacher Distillation

We present a novel Task-Specific Teacher Distillation training paradigm to train the *StatsMerge-Learner* (SML) for model merging as illustrated in Fig. 2 and detailed in Algorithm 1. Our key intuition is that each pre-trained model $\theta_k$ is already good at its own task dataset $\{x_i, y_i\}_k \in D_k$, therefore we regard it ($\theta_k$) as the Task-Specific Teacher $T_k$. Subsequently, the predictions $\hat{y}_{i,k}$ from the model trained on task $k$ is decent enough as pseudo labels when it comes to its pre-trained dataset sample $\{x_i, y_i\}_k$. We aggregate such pairs $\{x_i, \hat{y}_{i,k}\}_k$ to construct the merged dataset to train $SML()$. We highlight the key benefit of this approach that enables dataset preparation without relying on human-annotated labels. The predicted class label in one-hot encoded format. Therefore, the cross-entropy loss is applied:

$$\mathcal{L}_{\text{CE}} = -\sum_{c=1}^{C_m} \hat{y}_{c,k} \log(\hat{y}_c)) \tag{7}$$

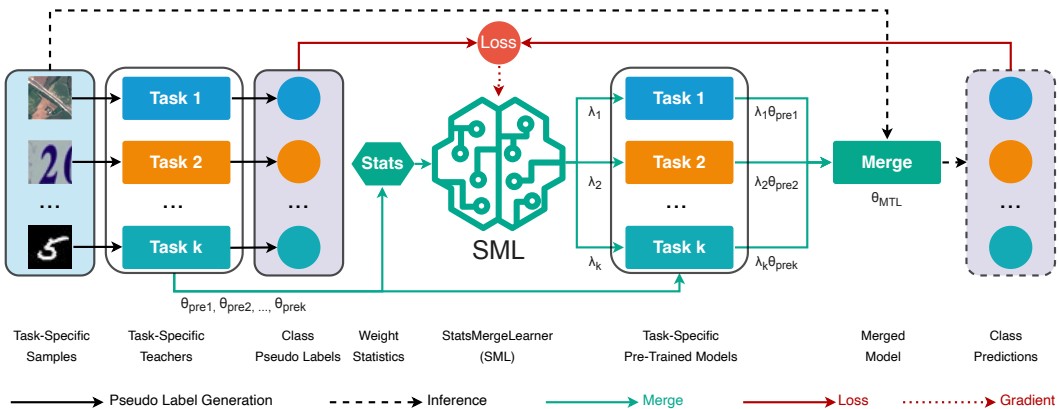

Figure 2: Knowledge Distillation Diagram. *StatsMergeLearner* (SML) learns the merging coefficients $\lambda$ by minimizing the loss between the merged model's predictions and pseudo labels generated by task-specific teacher models. During inference, only the merged model in *StatsMerging* is used to predict class labels.

We present a novel Task-Specific Teacher Distillation training paradigm to train the *StatsMergeLearner* (SML) for model merging as illustrated in Fig. 2 and detailed in Algorithm 1. Our key intuition is that each pre-trained model $\theta_k$ is already good at its own task dataset $\{x_i, y_i\}_k \in D_k$, therefore we regard it ($\theta_k$) as the Task-Specific Teacher $T_k$. Subsequently, the predictions $\hat{y}_{i,k}$ from the model trained on task $k$ is decent enough as pseudo labels when it comes to its pre-trained dataset sample $\{x_i, y_i\}_k$. We aggregate such pairs $\{x_i, \hat{y}_{i,k}\}_k$ to construct the merged dataset to train $SML()$. We highlight the key benefit of this approach that enables dataset preparation without relying on human-annotated labels. The predicted class label in one-hot encoded format. Therefore, the cross-entropy loss is applied while such loss function simplicity helps extend to other vision tasks and architectures.

**Algorithm 1.** Unified Statistics-Guided Model Merging via Task-Specific Teacher Model Distillation

1: **Input:** Set of pre-trained models $\{M_1, M_2, \ldots, M_k\}$ with weights $\{\theta_1, \theta_2, \ldots, \theta_k\}$ for $K$ tasks.
2: **Output:** Merged model $M_{\text{merged}}$ with weights $\theta_{\text{merged}}$
3: // Prepare $K$ pre-trained models
4: **if** Same architecture $A$ for all $M_i$ **then**
5:      Set $M_{\text{target}}$ to the shared architecture
6: **else**
7:      Select a target architecture $M_{\text{target}}$
8:      **for** $i = 1$ to $k$ **do**
9:          **if** $A(M_i) \neq A(M_{\text{target}})$ **then**
10:              Distill $M_i$ into $M_{\text{target}}$ to obtain updated $\theta_i$
11:          **end if**
12:      **end for**
13: **end if**
14: // Merge $K$ models
15: **for** $k = 1$ to $K$ **do**
16:      // mean $\mu$, std $\sigma^2$, norm $m$, singular value $\sigma'_r$
17:      Extract statistics $S_k = [\mu, \sigma^2, m, \sigma'_r]$ from $\theta_k$
18:      Predict coefficients $\lambda_k = \text{SML}(S_k)$
19:      Merge layer weights: $\theta^l_{\text{merged}} = \sum_{i=1}^{k} \lambda_k \theta_k$
20: **end for**
21: **return** $M_{\text{merged}}$ with weights $\theta_{\text{merged}}$

# 4 Experiments

## 4.1 Experimental Setup

In this section, we present the experimental setup and evaluation results used to compare our method against recent baselines.

**Datasets and Models** : Our experiments include eight image classification tasks with datasets SUN397 (Xiao et al., 2016), Stanford Cars (Krause et al., 2013), RESISC45 (Cheng et al., 2017), EuroSAT (Helber et al., 2019), SVHN (Netzer et al., 2011), GTSRB (Stallkamp et al., 2011), MNIST (LeCun et al., 1998), DTD (Cimpoi et al., 2014), and CIFAR10 (Krizhevsky and Hinton, 2009) [2] We use ViT-B/32 CLIP (Radford et al., 2021) as the pre-trained backbone. Individual task-specific models are obtained by training on each dataset separately. For merging models with different architectures, we first distill them into a single backbone before applying our merging method.

---

[2]In the remainder of the paper, the abbreviations shown in brackets are used to denote each task dataset: SUN397 (SU), Cars (CA), RESISC45 (RE), EuroSAT (EU), SVHN (SV), GTSRB (GT), MNIST (MN) and DTD (DT).

**Baselines and Metrics** : We compare against standard baselines including Individual Training, Traditional Multi-Task Learning (MTL) (Zhang and Yang, 2021), Weight Averaging (Wortsman et al., 2022), Task Arithmetic (Ilharco et al., 2023), Fisher Merging (Matena and Raffel, 2022), RegMean (Jin et al., 2023), Ties-Merging (Yadav et al., 2023a) and AdaMerging (Yang et al.). The primary evaluation metric is the average accuracy (Avg Acc) on the test sets of all tasks. The evaluation is conducted on eight different vision classification tasks.

*StatsMergeLearner* **Training Detail** : Our MLP-based *StatsMergeLearner* learns to predict layer-wise or task-wise merging weights coefficients ($\lambda$) based on weight statistics from individual task models. The *StatsMergeLearner* is trained for 500 epochs using Adam, with a learning rate of $1e - 3$ and a StepLR scheduler (factor 0.1 every 100 epochs), which translates to around only 3 hours to merge 4 ViTs, offering the practicality and advantage of applying our technique for practitioners without spending days or weeks for training (Zhang and Yang, 2021; Padmanabhan et al., 2023). We train the *StatsMergeLearner* primarily using knowledge distillation from the aggregated dataset without human annotated labels described in Sec. 3.3, optimized with either Cross-Entropy (Mao et al., 2023) or KL Divergence (Kullback and Leibler, 1951) loss.

## 4.2 Merging Performance

In this section, we present a comprehensive evaluation of our approach in comparison to state-of-the-art task vector merging methods, assessing its superiority across several fundamental aspects: Multi-task merging performance, generalization to unseen tasks and heterogeneous architectures.

**Substantially Higher Merging Performance.** The main results of merging performance of ViT-B/32 models on eight tasks are presented in this section, detailed [3] in Table 2. We present two levels of granularity: Task-Wise (TW) and Layer-Wise (LW). Our method *StatsMerging* achieved an average accuracy (Avg Acc) of $76.5\%$ and $84.5\%$ in both TW and LW levels, outperforming the state-of-the-art (SOTA) method AdaMerging++ by a large margin of $3.3\%$ and $3.4\%$. We attribute the improvements to the ability of *StatsMergeLearner* to adapt task-specific weights based on their weight statistics to the merged model. The use of pseudo labels from task-specific teachers $\{T_1, T_2, \ldots, T_k\}$ provides stronger signals for *StatsMergeLearner* to better assign weight coefficients $\lambda$ than the entropy minimization approach in the AdaMerging++.

Table 2: Multi-task merging performance (Avg Acc %) when merging ViT-B/32 models on eight tasks. Results of our method *StatsMerging* are shaded in gray. Bold and underscore indicate the highest and second-highest scores within the merging group below the double rules in each column, respectively. TW: Task-wise. LW: Layer-wise.

| Method | SU | CA | RE | EU | SV | GT | MN | DT | Avg Acc |
|---|---|---|---|---|---|---|---|---|---|
| Pre-Trained | 62.3 | 59.7 | 60.7 | 45.5 | 31.4 | 32.6 | 48.5 | 43.8 | 48.0 |
| Individual | 75.3 | 77.7 | 96.1 | 99.7 | 97.5 | 98.7 | 99.7 | 79.4 | 90.5 |
| Traditional MTL | 73.9 | 74.4 | 93.9 | 98.2 | 95.8 | 98.9 | 99.5 | 77.9 | 88.9 |
| Weight Averaging | 65.3 | 63.4 | 71.4 | 71.7 | 64.2 | 52.8 | 87.5 | 50.1 | 65.8 |
| Task Arithmetic | 55.2 | 54.9 | 66.7 | 78.9 | 80.2 | 69.7 | 97.3 | 50.4 | 69.1 |
| Fisher Merging | **68.6** | 69.2 | 70.7 | 66.4 | 72.9 | 51.1 | 87.9 | 59.9 | 68.3 |
| RegMean | 65.3 | 63.5 | 75.6 | 78.6 | 78.1 | 67.4 | 93.7 | 52.0 | 71.8 |
| Ties-Merging | 59.8 | 58.6 | 70.7 | 79.7 | 86.2 | 72.1 | **98.3** | 54.2 | 72.4 |
| TW AdaMerging | 58.0 | 53.2 | 68.8 | 85.7 | 81.1 | 84.4 | 92.4 | 44.8 | 71.1 |
| TW AdaMerging++ | 60.8 | 56.9 | 73.1 | 83.4 | 87.3 | 82.4 | 95.7 | 50.1 | 73.7 |
| **TW *StatsMerging*** | 61.3 | 70.0 | 74.2 | 85.2 | 87.5 | 82.5 | 96.2 | 54.2 | 76.4 (+3.3) |
| LW AdaMerging | 64.5 | 68.1 | 79.2 | 93.8 | 87.0 | 91.9 | 97.5 | 59.1 | 80.1 |
| LW AdaMerging++ | 66.6 | 68.3 | 82.2 | **94.2** | 89.6 | 89.0 | **98.3** | 60.6 | 81.1 |
| **LW *StatsMerging*** | 67.4 | **74.1** | **82.9** | 91.1 | **89.8** | **94.7** | **98.3** | **77.5** | **84.5 (+3.4)** |

The LW *StatsMerging* achieved significantly higher ($+8.1\%$) Avg Acc than TW *StatsMerging* with $84.5\%$ and $76.4\%$, respectively. This improvement in layer-wise merging over task-wise aligns with observations in AdaMerging. We hypothesize that compared to the coarser task-level granularity, the

---

[3]Please refer to the Appendix for experimental details, including the full list of tasks, datasets, and baselines.

finer-grained layer-level offers greater flexibility of coefficients across various semantics regarding task-agnostic and task-specific features, which are often learned in various levels of a neural network.

**Significantly Enhanced Generalization.** A merged model is expected to generalize to unseen tasks by strategically transferring the knowledge from the combined set of old tasks. We benchmark such generalization ability of *StatsMerging* against four strong baselines: Task Arithmetic, Ties-Merging, AdaMerging, and AdaMerging++. We follow the same evaluation protocol in AdaMerging training on two groups of tasks, each group consisting of six seen tasks, and testing on two unseen tasks.

Table 3: Generalization results (Avg Acc %) on two unseen tasks when merging Layer-Wise ViT-B/32 models on six tasks. *StatsMerging*: shaded in gray. Bold: top score. Underscore: 2nd-highest score.

| | Seen Tasks | | | | | | | | Unseen Tasks | | |
|---|---|---|---|---|---|---|---|---|---|---|---|
| **Method** | SU | CA | RE | DT | SV | GT | **Avg Acc** | | MN | EU | **Avg Acc** |
| Task Arithmetic | 63.3 | 62.4 | 75.1 | 57.8 | 84.6 | 80.4 | 70.6 | | 77.2 | 46.2 | 61.7 |
| Ties-Merging | 67.8 | 66.2 | 77.2 | 56.7 | 77.1 | 70.9 | 69.3 | | 75.9 | 43.3 | 59.6 |
| AdaMerging | 65.2 | 65.9 | **88.5** | 61.1 | 92.2 | _91.5_ | 77.4 | | _84.0_ | _56.1_ | _70.0_ |
| AdaMerging++ | _68.2_ | 67.6 | 86.3 | _63.6_ | _92.6_ | 89.8 | _78.0_ | | 83.9 | 53.5 | 68.7 |
| *StatsMerging* | **69.1** | **71.3** | _86.7_ | **75.2** | **93.2** | **95.7** | **81.9 (+3.9)** | | **85.1** | **56.4** | **70.8 (+0.8)** |
| **Method** | SU | CA | GT | EU | DT | MN | **Avg Acc** | | RE | SV | **Avg Acc** |
| Task Arithmetic | 64.0 | 64.0 | 75.2 | 87.7 | 57.0 | 95.7 | 73.9 | | 52.3 | 44.9 | 51.1 |
| Ties-Merging | 68.0 | 67.1 | 67.7 | 78.4 | 56.5 | 92.8 | 71.8 | | **58.7** | 49.2 | 53.9 |
| AdaMerging | 67.1 | 67.8 | _94.8_ | _94.4_ | 59.6 | 98.2 | 80.3 | | 50.2 | 60.9 | 55.5 |
| AdaMerging++ | _68.9_ | _69.6_ | 91.6 | 94.3 | _61.9_ | _98.7_ | _80.8_ | | 52.0 | 64.9 | _58.5_ |
| *StatsMerging* | **69.6** | **73.3** | **96.1** | **95.4** | **74.1** | **97.2** | **84.3 (+3.5)** | | _54.2_ | **67.1** | **60.7 (+2.2)** |

Details are presented in Table 3, where in both groups our proposed *StatsMerging* achieved $70.8\%s$ and $60.7\%$, significantly outperforming the second best method AdaMerging by $+0.8\%$ and $+2.2\%$ margins. Such improvements are attributed to both (1) the careful feature design of weight statistics that captures the dominant information regarding weight distributions from pre-trained models, which potentially helps reduce noise from each task dataset; and (2) the joint training from all old tasks on the task-specific teacher-distilled labels, enabling the implicit learning of task-agnostic and task-specific features that can benefit the generalization ability.

**Extension to Heterogeneous Architectures.** Our *StatsMerging* offers the first and unique advantage without the assumption of architectural identity in prior works (Wortsman et al., 2022; Ilharco et al., 2023; Yadav et al., 2023a; Matena and Raffel, 2022; Jin et al., 2023). To verify the performance of varying architectures, we conduct experiments on ResNet50 (RN) and ViT-B/32 (VT) to represent Convolutional Neural Network (CNN) and Vision Transformer (ViT) architectures.

In particular, we distill fine-tuned VT teachers into a RN (Khanuja et al., 2021) student on three diverse tasks of CIFAR-10 (CI), EuroSAT (EU) and Stanford Cars (CA) with the distillation loss:

$$\mathcal{L} = \alpha \mathcal{L}_{\text{CE}}(y, \hat{y}) + (1 - \alpha) T^2 \mathcal{L}_{\text{KL}}\big(\sigma(\tfrac{z}{T}), \sigma(\tfrac{z_t}{T})\big),$$
(8)

where $\mathcal{L}_{\text{KL}}$ denotes KL-Divergence, $z$ is logit, $T = 4.0$ represents temperature, $\alpha = 0.7$ is the weight balance of two sub-losses. CI is used due to the available pre-trained RN weights. Remarkably, the distilled RN matches its VT teacher's accuracy, achieving $76.4\%$ (VT: $77.7\%$) for CA and $94.5\%$ for EU (VT: $99.7\%$) despite the architectural difference shown in Table 3. We then apply our *StatsMerging* to combine the CI–trained RN and its distilled variants. We merge multiple task models into a single RN using the merging coefficients inferred by *StatsMergeLearner*, yielding a $7.6\%$ average improvement over the vanilla Task-Arithmetic of $73.7\%$ and achieving $81.3\%$ average accuracy.

Table 3. Multi-task merging performance (Avg Acc %) of models in heterogeneous architectures: ResNet50 (RN) & ViT-B/32 (VT). *StatsMerging*: shaded in gray.

| **Method** | CI | CA | EU | **Avg Acc** |
|---|---|---|---|---|
| Backbone Distilled | RN - | VI RN | VI RN | - - |
| Individual Distilled | 97.8 - | 77.7 76.4 | 99.7 94.5 | 91.7 - |
| Weight Averaging | 77.1 | 56.4 | 64.9 | 59.4 |
| Ties-Merging | 76.5 | 52.8 | 80.1 | 69.8 |
| Task Arithmetic | 81.4 | 61.6 | 78.2 | 73.7 |
| **LW *StatsMerging*** | 87.2 | 68.4 | 88.4 | 81.3 |

### 4.3  *StatsMerging* Analysis

**Label & Loss Function Study.**
We conduct a loss function study on ViT-B/32 (4) models merged from four tasks, as shown in Table 4. Observe that *StatsMerging* trained on pseudo labels via Task-Specific Teacher Distillation (KD) achieves similar performance to *StatsMerging* trained on ground-truth labels (GT), with $88.5\%$ and $81.2\%$ average accuracy in TW and $90.4\%$ and $83.5\%$ in LW levels.

Table 4. Multi-task performance (Avg Acc %) of *StatsMerging* when merging ViT-B/32 (4) models across four tasks. *StatsMerging* shaded in gray. GT: Ground Truth. KD: Knowledge Distillation. TW: Task-wise. LW: Layer-wise.

| Loss | Level | CA | EU | RE | GT | Avg Acc |
|------|-------|------|------|------|------|---------|
| GT | TW | 73.2 | 94.2 | 91.1 | 95.6 | 88.5 |
| KD | TW | 64.2 | 88.6 | 85.2 | 86.7 | 81.2 |
| GT | LW | 75.6 | 96.3 | 92.1 | 97.6 | 90.4 |
| KD | LW | 68.7 | 91.6 | 87.2 | 93.5 | 83.5 |

**Statistical Feature Ablation Study.**
We conduct an ablation study on the statistical features. Results in Table 5 show that combining all statistical features improves merging performance, validating our design choice. Notably, the singular values $\sigma'$ improve the multi-task performance in both same and different architecture settings by $+3.0$ and $+3.2$ increase of average accuracy, justifying our design choice of using SVD.

Table 5: Multi-task performance (Avg Acc %) of *StatsMerging* when ablating statistical features of ViT-B/32 (4) models on four tasks: CA, EU, RE & GT. Bold: top score. *StatsMerging*: shaded in gray.

| \multicolumn Same Architecture | | | | | Different Architecture | | | | |
|-----------|-----------|-----|-----------|-----------|-----------|-----------|-----|-----------|-----------|
| $\mu_{\theta_k}$ | $\sigma^2$ | $m$ | $\sigma'$ | Avg Acc | $\mu_{\theta_k}$ | $\sigma^2$ | $m$ | $\sigma'$ | Avg Acc |
| ✓ | | | | 83.4 | ✓ | | | | 76.2 |
| ✓ | ✓ | | | 84.1 (+0.7) | ✓ | ✓ | | | 77.5 (+1.3) |
| ✓ | ✓ | ✓ | | 87.2 (+3.1) | ✓ | ✓ | ✓ | | 78.1 (+0.6) |
| ✓ | ✓ | ✓ | ✓ | **90.2 (+3.0)** | ✓ | ✓ | ✓ | ✓ | **81.3 (+3.2)** |

**Coefficient Analysis.** We visualize the heatmap of ViT-B/32 (4) across eight tasks in Fig. 3. We make several key observations: (1) the **common recurring pattern** of coefficients $\lambda$ across all eight tasks from earlier (left) to deeper (right) layers aligns with the repeated self-attention blocks in the ViT architecture, e.g. Multi-Head Self-Attention (MHSA), MLP (Feed-Forward Network), and LayerNorm, etc, demonstrating the need of various coefficients for various types of layers; (2) The **sparse non-uniform coefficient distributions** (various colors like Layer 13, 19 or 25) suggests that merging layers can be more efficient at some specific layers instead of using one coefficient for an entire pre-trained model, justifying the our granularity choice of Layer-Wise over Task-Wise level; (3) some **task-specific coefficient distributions** verify the necessity of assigning distinct merging coefficients across tasks in various layers, such as in Layer 5 vs. 147. Such distributions reflect the various visual representations for different semantics learned across both layers and tasks. More visualizations are provided in the Appendix for in-depth analysis.

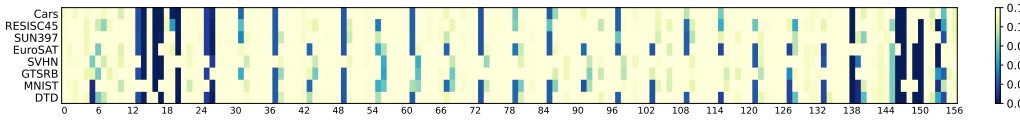

Figure 3: Heatmap of *StatsMerging* merging coefficients $\lambda$ of ViT-B/32 (4) across eight tasks. X-axis: layer index. Y-axis: Tasks. Coefficients are normalized to sum to 1.

## 5  Conclusion

Model merging offers a compelling post-hoc advantage to reduce memory storage from a corpus of large pre-trained models. We propose *StatsMerging*, a novel merging technique without relying on simple heuristics, test-time samples or human annotated. The key intuition lies in the guidance of weight statistics using a lightweight MLP learner, dubbed *StatsMergeLearner*, to learn merging coefficient prediction. Exhaustive experiments demonstrate the effectiveness of our proposed *StatsMerging* in model mering from eight diverse tasks, achieving $84.5\%$ average accuracy and surpassing the SOTA AdaMerging ($81.1\%$) by a large margin of $3.4\%$.

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
