# OpenReview forum: "StatsMerging: Statistics-Guided Model Merging via Task-Specific Teacher Distillation"
_NeurIPS.cc/2025/Conference — Submitted to NeurIPS 2025_

### Official Review · Reviewer_hVS2 · 2025-06-02

**Clarity:** 2
**Significance:** 3
**Originality:** 3
**Rating:** 3
**Confidence:** 4

**Summary:**

The authors present StatsMerging, a novel approach that leverages singular values from singular value decomposition (SVD) to capture task-specific weight distributions, which serve as proxies for task importance in guiding the learning of task coefficients. They introduce a lightweight component, StatsMergeLearner, designed to model the weight distributions of task-specific pre-trained models efficiently. The method incorporates two forms of knowledge distillation: (a) distilling knowledge from task-specific models to train the StatsMergeLearner, and (b) for the first time, distilling knowledge from models with different architectures prior to merging, following a distill-then-merge paradigm. Extensive experiments across eight tasks validate the effectiveness of StatsMerging.

**Questions:**

1. In Figure 1(d), the notation *stats(θ\_pre)* appears to be inaccurate—it should be *stats(θ\_k)* to correctly refer to task-specific parameters.
2. The term *test-time adaptability* is unclear. Could the authors clarify what this entails in the context of model merging?
3. The method uses rank3 singular value from the singular value vector. Have the authors experimented with other ranks to assess sensitivity to this choice?
4. What is the rationale behind using singular values to capture task-specific characteristics? A theoretical or empirical justification would strengthen the argument.
5. How is the aggregated dataset constructed? Does it originate from the training set, the test set, or an external data source?

**Ethical Concerns:**

["NO or VERY MINOR ethics concerns only"]

**Final Justification:**

I consider rebuttal and discussions with authors and other reviewers.

**Limitations:**

They did mention limitations in the appendix.

**Paper Formatting Concerns:**

The references section should appear before the Checklist.

**Quality:**

2

**Strengths And Weaknesses:**

**Strengths**

1. The authors propose *StatsMerging*, a novel and lightweight learning-based model merging method that relies on weight distribution statistics, requiring neither ground-truth labels nor test samples during training.
2. They design the *StatsMergeLearner*, a lightweight module that estimates merging coefficients (λ) using statistical features of model weights, trained via a newly introduced Task-Specific Teacher Distillation paradigm.
3. Extensive experiments across multiple tasks demonstrate the effectiveness and generalizability of StatsMerging.

**Weaknesses**

1. Although Eq. (7) claims to use pseudo-labels and avoid human-annotated labels, Eq. (8) relies on the labeled aggregated dataset during distillation. This raises questions about the true advantage of avoiding labeled data.
2. The training of StatsMergeLearner requires generating a merged model and performing inference in each iteration, which may significantly increase training time. Has this overhead been quantified?
3. Algorithm 1 refers to "Distill Mi into M\_target to obtain updated θ\_i", but this step is not explained in the Methodology section, making it unclear and abrupt.
4. Could existing methods also support heterogeneous architectures if combined with a distillation process? The paper does not sufficiently compare with such potential alternatives.

---

> ### Author Rebuttal · Authors · 2025-07-29
>
> We sincerely thank Reviewer __hVS2__ (R4) for recognizing the __novelty__ and __lightweight__ design of our proposed approach, as well as the __effectiveness__ and __generalizability__ of the method supported by __extensive experiments conducted across multiple tasks__. The following responses address each concern individually, with __AR4W#__ denoting weaknesses and __AR4Q#__ denoting questions. A: Answer.
> ___
> R4W1: *Clarity on pseudo-label notations in Eq. (8) during distillation.*
>
> __AR4W1__: Only __pseudo labels__ are used during distillation in Eq. (8) in our implementation. Eq. (8) will be updated with the following equation to avoid confusion in the camera-ready version:
>
> $L=αL_{CE}(\hat{y_{k}}, \hat{y}) + (1-α)T^{2}L_{KL}(\sigma(\frac{z}{T}), \sigma(\frac{z_{t}}{T}))$ (8)
>
> where $\hat{y_{k}}$ refers to the predictions of the teacher for task $k$ as __pseudo labels__ and $\hat{y}$ is the predictions of the merged model.
>
> We would like to thank the reviewer's comment and the opportunity to enhance clarity of this work.
> ___
> R4W2: *Training overhead of generating a merged model and performing inference in each iteration.*
>
> __AR4W2__: In each epoch, *SML* takes 0.0338s while *merge+load" takes 2.5534s. Merging happens in every 100 epochs. We will append this detail to Appendix in the camera-ready version.
>
> Please also refer to __AR3W4__ & __AR3Q4__ for inference cost analysis: In Table 4 in Sec. B.4 Efficient Inference in the Appendix in the original submission.
>
> StatsMergeLearner itself is orders of magnitude smaller and computationally lighter than the merged model, with only 0.336M parameters, 0.73M MACs and 1.46M FLOPs on an NVIDIA RTX A6000 GPU. The results demonstrate that StatsMergeLearner introduces __negligible overhead__ in terms of parameters (SML-to-Merged Model __Ratio__: 0.336M / 10.99M = __0.0306__) and computation (SML-to-Merged Model __Ratio__: 1.46M / 2.95G = __0.0005__).
> ___
> R4W3: *Algorithm 1 refers to "Distill Mi into M_target to obtain updated θ_i", but this step is not explained in the Methodology section, making it unclear and abrupt.*
>
> __AR4W3__: Due to the page limit, we intended to retain the key information in main text in Sec. 3 Methodology and illustrate the details in the algorithm or Appendix if needed. We present a concrete experiment for __this step with explanations__ in Sec. Extension to Heterogeneous Architectures where we distill fine-tuned ViT teachers into a ResNet student in Line 255 with the distillation loss defined in Eq. (8).
>
> We thank the reviewer for the valuable comment and will append the description and explanation of Step 10 of Algorithm 1 "Distill $M_{i}$ into $M_{target}$ to obtain updated $θ_{i}$" to Line 193 in Sec. 3 Methodology. Specifically the following text will be added in the camera-ready version: "When the current ($M_{i}$) and target models' ($M_{target}$) architectures are different, we first initialize a model that shares the same architecture as $M_{target}$ which will be treated as the student model. Then on the dataset where the current ($M_{i}$) model was trained on, we feed samples into $M_{i}$ to produce pseudo labels to train $M_{target}$ until converged. In this way, models from all tasks will share the same architecture with $M_{target}$ for the subsequent merging."
> ___
> R4W4: *Support heterogeneous architectures if combined with a distillation process.*
>
> __AR4W4__: Please refer to __AR2W2__.
> ___
> R4Q1: *In Figure 1(d), the notation stats(θ_pre) appears to be inaccurate—it should be stats(θ_k) to correctly refer to task-specific parameters.*
>
> __AR4Q1__: Figure 1 is the __high level conceptual__ illustration figure that summarizes the key differences between our proposed approach (d) vs previous methods (a)-(c) while the underlying details (such as the specific notations) are presented in (AR4Q1a) Sec. 3.2 Weight Statistics-Guided Model Merging and (AR4Q1b) implementation in the Github link in the submission. Specifically, in AR4Q1a and AR4Q1b, statistical features are computed from a task vector $\theta_{k}$ in the following way: $stats(\theta_{k})$. Therefore, we agree with the reviewer on using $stats(\theta_k)$ in Figure 1 (d) to better align with our actual implementation. We thank the reviewer for this helpful comment to further improve the paper's clarity and presentation.
> ___
> R4Q2: *Explanation of the term test-time adaptability*
>
> __AR4Q2__: Test-time adaptability refers to the ability of a model to adjust its weights to unseen data during inference without access to human-labeled annotations. This characteristic is mainly inspired from AdaMerging (Yang et al.) and prior works [1-4].
>
> We will incorporate the explanation into the caption of Table 1. We appreciate the reviewer’s valuable comment, which helps improve the presentation of our paper.
>
> [1] Dequan Wang, Evan Shelhamer, Shaoteng Liu, Bruno Olshausen, and Trevor Darrell. Tent: Fully test-time adaptation by entropy minimization. In ICLR, 2021.
>
> [2] Shuaicheng Niu, Jiaxiang Wu, Yifan Zhang, Zhiquan Wen, Yaofo Chen, Peilin Zhao, and Mingkui Tan. Towards stable test-time adaptation in dynamic wild world. In ICLR, 2023.
>
> [3] Yves Grandvalet and Yoshua Bengio. Semi-supervised learning by entropy minimization. NeurIPS, 17, 2004.
>
> [4] Subhankar Roy, Martin Trapp, Andrea Pilzer, Juho Kannala, Nicu Sebe, Elisa Ricci, and Arno Solin. Uncertainty-guided source-free domain adaptation. In ECCV, pp. 537–555. Springer, 2022.
>
> ___
> R4Q3: *Justification of the choice of rank of singular values.*
>
> __AR4Q3__: Please refer to __AR1Q2.2__.
>
> ___
> R4Q4.1: *Rationale behind using singular values to capture task-specific characteristics? A theoretical justification.*
>
> __AR4Q4.1__: Please refer to __AR3W1 & AR3Q1__.
>
> R4Q4.2: *Rationale behind using singular values to capture task-specific characteristics? An empirical justification.*
>
> __AR4Q4.2__: As shown in Table 5 of the __original submission__, adding singular values (__σ′__) to the set of statistical features generally leads to the __most significant performance improvement__ in both the Same Arch (+3.0) and Different Arch (+3.2) settings (Arch: Architecture). This provides empirical evidence to support the design choice of incorporating singular values. For your convenience, we include Table 5 here:
>
> **Table 5**: Multi-task performance (Avg Acc %) of *StatsMerging* when ablating statistical features of ViT-B/32 (4 models) on four tasks: CA, EU, RE & GT. **Bold**: top score.
>
> |||**Same**|**Arch**|||||**Different**|**Arch**|||||
> |-|-|:-:|:-:|:-:|:-:|:-:|:-:|:-:|:-:|:-:|:-:|:-:|:-:|
> ||μ|σ²|m|σ′|**Avg Acc**||μ|σ²|m|σ′|**Avg Acc**|
> ||✓||||83.4||✓||||76.2|
> ||✓|✓|||84.1 (+0.7)||✓|✓|||77.5 (+1.3)|
> ||✓|✓|✓||87.2 (+3.1)||✓|✓|✓||78.1 (+0.6)|
> |__SML (Ours)__|✓|✓|✓|✓|**90.2 (+3.0)**||✓|✓|✓|✓|**81.3 (+3.2)**|
> ___
> R4Q5: *How is the aggregated dataset constructed? Does it originate from the training set, the test set, or an external data source?*
>
> __AR4Q5__: It originates from the training/val/test sets from the K tasks. We will add the details in Appendix in the camera-ready version.
>
> Lastly, we sincerely thank the reviewer for their time and effort in providing invaluable feedback. __ALL__ of the comments, including the related works mentioned by the reviewer, will be incorporated and reflected in the __camera-ready__ version. We welcome and encourage questions focused on the key contributions we have claimed: exploiting singular values from __SVD__, the __first heterogeneous architectural merging__ via distillation __without human-annotated labels__, and extensive empirical validation.

---

> > ### Comment · Reviewer_hVS2 · 2025-08-03
> >
> > The authors' response has addressed some of my concerns. Regarding AR4Q5, could you clarify what is meant by "It originates from the training/val/test sets from the K tasks"? Does this imply that the sets are combined? Additionally, I believe that the performance improvement may still primarily benefit from teacher distillation.

---

> > > ### Author Response · Authors · 2025-08-03
> > > **Details of AR4Q5 in Progress**
> > >
> > > __Yes__.
> > >
> > > We thank Reviewer hVS2 (R4)'s time and efforts in responding to our responses. Acknowledged the request of the details of __AR4Q5__. We will provide the details formally shortly. Please stay tuned. We thank Reviewer hVS2 for identifying the clarity gap and helping to further improve the presentation of our work.

---

> > > ### Author Response · Authors · 2025-08-06
> > >
> > > Dear Reviewer hVS2 (R4),
> > >
> > > We hope our responses address your concerns. Please feel free to reach out if anything remains unclear. Thank you!

---

> ### Author Response · Authors · 2025-08-05
> **Extended Details of Dataset Aggregation AR4Q5**
>
> The way datasets are aggregated was stated on Line 193 in the __initial submission__. We quote the description for your convenience:
>
> *"Our key intuition is that each pre-trained model $\theta_{k}$ is already good at its own task dataset {$x_{i}$, $y_{i}$}$\_{k}$ $\in$ $D_{k}$, therefore we regard it ($\theta_{k}$) as the Task-Specific Teacher $T_{k}$. Subsequently, the predictions $\hat{y}\_{i, k}$ from the model trained on task $k$ serves as sufficiently reliable pseudo labels for the validation dataset sample {$x_{i}$, $y_{i}$}$\_{k}$ from the same task. We aggregate such pairs {$x_{i}$, $\hat{y}\_{i,k}$}$\_{k}$ to construct the merged dataset to train $SML()$."*
> ___
> We extend the detail as follows:
>
> __(1) Task-Specific Teacher Models Preparation.__
>
> Collect $K$ pre-trained models $\Theta$ = {$\theta_{1}$, $\theta_{2}$, ..., $\theta_{k}$} while each model weight is fine-tuned on an independent task $k$ on dataset {$x_{i}$, $y_{i}$}$\_{k}$ $\in$ $D_{k}$, where $D_{k}$ denotes the dataset for task $k$, $x_{i}$ and $y_{i}$ represent a sample's input and its corresponding . Note that $y_{i}$ is not use for $SML$ learning but only used in the evaluation step.
>
> __(2) Train/Val/Test Split.__
>
> Each dataset $D_{k}$ for task $k$ is split into training, validation and test sets with an 8:1:1 ratio, denoted as $D_{k}^{train}$, $D_{k}^{val}$ and $D_{k}^{test}$, respectively.
>
> __(3) Pseudo Label Preparation for Training Set $D\^{train}$.__
> Following (2), for task $k$, the task-specific teacher $\theta_{k}$ takes a sample $x_{i,k}$ and generate its prediction $\hat{y}\_{i,k}$ as a pseudo label. The resulting pairs of ($x_{i,k}$, $\hat{y}\_{i,k}$) are aggregated to form task $k$'s training dataset $D\^{train}\_{k}$ $\subseteq$ $D\^{train}$.
>
> __(4) Val $D\^{val}$ and Test Set$D\^{test}$ Preparation.__
> Following (2), for task $k$, the original pairs ($x_{i,k}$, $y\_{i,k}$) in the split validation set ($D\^{val}\_{k}$ $\subseteq$ $D\^{val}$) or test set ($D\^{test}\_{k}$ $\subseteq$ $D\^{test}$) are used, where $y\_{i,k}$ is the human-annotated ground truth label used solely for evaluation.
>
> __(5) Complete Dataset $D$ Preparation.__
> Aggregate $D\^{train}$, $D\^{val}$, and $D\^{test}$ to form the complete dataset $D$ = {$D\^{train}$, $D\^{val}$, $D\^{test}$}.
>
> Concretely, in the eight vision tasks, the samples {$x_{i}$, $y_{i}$}$\_{k}$ $\in$ $D_{k}$ are drawn from the following datasets SUN397 (SU), Cars (CA), RESISC45 (RE), EuroSAT (EU), SVHN (SV), GTSRB (GT), MNIST (MN) and DTD (DT), while the pseudo label $\hat{y}\_{i}$ is generated by the Task-Specific Teacher set $\Theta$ = {$\theta_{1}$, $\theta_{2}$, ..., $\theta_{k}$}. These are aggregated to constitute the overall dataset $D$.
> ___
>
> We will include the extended details in the appendix of the camera-ready version. Please let us know if you have any further questions.

---

> ### Author Response · Authors · 2025-08-05
> **Effect of Teacher Distillation**
>
> Regarding this follow-up question:
>
> __R4Q6__: *I believe that the performance improvement may still primarily benefit from teacher distillation.*
>
> __AR4Q6__: We would like to clarify that teacher distillation in this model merging setting is mainly proposed as a method to avoid the use of human annotated ground truth (GT) labels (without human supervision).
>
> We follow the same argument with AdaMerging that a model merging method trained on __GT__ is treated as __upper-bound__ performance.
>
> In Table 4 of Sec. 4.3 StatsMergingAnalysis in the __initial submission__, we show that while our proposed method (trained without GT) achieves strong performance with over $80$%+ Avg Acc, there remains a gap (GAP) compared to the supervised method (trained on GT), which serves as the upper bound. These results suggest that training on ground truth (GT) could potentially further improve our proposed SML. However, our focus is on advancing it toward a direction without human supervision.
>
> For your convenience, we quote Table 4 from our submission below:
>
> Table 4. Multi-task performance (Avg Acc %) of *StatsMerging* when merging ViT-B/32 (4) models across four tasks.
> GT: Ground Truth. KD: Knowledge Distillation. TW: Task-wise. LW: Layer-wise.
>
> | Loss | Level | CA   | EU   | RE   | GT   | Avg Acc |
> |------|-------|------|------|------|------|---------|
> | GT   | TW    | 73.2 | 94.2 | 91.1 | 95.6 | __88.5__    |
> | __KD__   | TW    | 64.2 | 88.6 | 85.2 | 86.7 | __81.2__    |
> | GT   | LW    | 75.6 | 96.3 | 92.1 | 97.6 | __90.4__    |
> | __KD__   | LW    | 68.7 | 91.6 | 87.2 | 93.5 | __83.5__    |
>
> *Update on August 7, 2025*:
>
> The GAP is also observed in Updated Table 2 __UAR1W2__, from which we highlight the key results in Table 2mini as follows:
>
> **Updated Table 2mini. Knowledge distillation improves model merging performance**
>
>
> | Method                   | CI   | CA   | EU   | Avg Acc |
> |--------------------------|------|------|------|--------------|
> | Multitask Learning (MTL)      | 96.4 | 74.6 | 96.2 | **89.1**         |
> | **Multitask Distilled (MTD)**      | 89.3 | 52.7  | 83.4 | **75.1**         |
>
>
>
>
> As stated in __AR3W1__ & __AR3Q1__, we attribute the performance gain to the compact representation from weight statistical distribution, especially SVG which captures the weight importance information and the dominant directions, as well as the SML's ability to learn merging coefficients for merging. Teacher distillation is primarily introduced to demonstrate that pseudo labels are sufficiently strong enough for effective SML training.

---

> ### Author Response · Authors · 2025-08-07
> **Feedback**
>
> Reviewer hVS2,
>
> We are grateful for your time and feedback. Our method has been acknowledged for its novelty, lightweight design, and demonstrated effectiveness and generalizability across diverse tasks. However, we respectfully find it difficult to align these acknowledgments and the reviews with the **initial score** of 2: Reject based on the description: "*For instance, a paper with technical flaws, weak evaluation, inadequate reproducibility and incompletely addressed ethical considerations.*"
>
> In contrast, our work:
>
> (a) Contains __no technical flaws__. Only minor clarifications were requested, which we have addressed in this rebuttal and will reflect in the camera-ready version.
>
> (b) Includes __extensive evaluation__. It covers experiments over 8 vision tasks and 6 NLP tasks, along with ablation studies, generalization and robustness analyses, an investigation into the impact model size, and noise tolerance analysis, etc. More experimental results are provided in the Appendix of the original submission.
>
> (c) Ensures __reproducibility__. Anonymized code is provided in the supplementary material and a GitHub link. Checkpoints will be released upon acceptance.
>
> (d) Poses __no ethical concerns__.
>
> We understand that the reviewer might have their own judgement, but we respectfully ask for consideration of whether the score aligns with the overall content and our rebuttal, especially given the fact that most of the review requests are about further clarity (__R4W1__, __R4W3__, __R4Q1__, __R4Q2__, __R4Q4__ & __R4Q5__). If possible, we respectfully ask the reviewer to provide specific and concrete reasons to support the given rating. We take all feedback seriously as an important opportunity to improve our work. Our team would be glad to address any further questions or concerns. We appreciate your time and consideration.

---

### Official Review · Reviewer_rPn2 · 2025-06-16

**Clarity:** 2
**Significance:** 3
**Originality:** 3
**Rating:** 3
**Confidence:** 3

**Summary:**

This paper presents a model merging approach. The authors propose a "Task-Specific Teacher Distillation" method for merging vision models with heterogeneous architectures. Compared to existing methods, the key advantage of this approach is that it operates without requiring ground truth labels or test samples.

**Questions:**

1 What is the rationale behind using SVD for feature space alignment? While the experiments demonstrate the method's effectiveness, providing additional theoretical insights or deeper analysis would further strengthen the contribution.

2 What is the fundamental difference between the proposed method and AdaMerging? If multiple tasks exhibit conflicting feature spaces, would this method still perform effectively?

3 There is a lack of theoretical analysis and verification.

4 Experimental results regarding inference efficiency are missing.

**Ethical Concerns:**

["NO or VERY MINOR ethics concerns only"]

**Final Justification:**

Thank you for your response, and I will keep my original recommendation.

**Limitations:**

see above

**Quality:**

3

**Strengths And Weaknesses:**

Strengths:

1 The research motivation is clearly articulated, and the experimental results effectively validate the method's efficacy.

2 The logic is coherent, and the paper is well-written and accessible.

Weaknesses:

1 The correlation between using SVD for feature space alignment remains unclear.

2 There is a lack of theoretical analysis.

3 The relationship between the method proposed by the authors and existing approaches, particularly AdaMerging, is not clearly defined.

4 Experimental analysis on efficiency is lacking.

---

> ### Author Rebuttal · Authors · 2025-07-29
>
> We thank Reviewer __rPn2__ (R3) for recognizing the __strength of our motivation, the coherent logical flow, the quality and accessibility of the presentation__, and the __effectiveness__ demonstrated by our experimental results. The following responses address each concern individually, with __AR3W#__ referring to weaknesses and __AR3Q#__ to questions. A: Answer.
> ___
> R3W1 & R3Q1: *Rationale behind using SVD for feature space alignment.*
>
> __AR3W1__ & __AR3Q1__: Based on the observations from TIES-Merging, there are two sources of interference for a merged model: __(a) redundant__ parameter values and __(b) sign conflict__ of a parameter. To address these issues, TIES-Merging employs a set of heuristics: Trim, Elect Sign, and Disjoint Merge.
>
> Since SVD decomposes a model's weight matrix $W$ into $U\Sigma V^{T}$, singular values in our case captures the __importance__ (via Sigma) for (a) with respect to its __dominant directions__ (via U and V) for (b). The top singular values yield a __compact representation__ of each model, reducing noice and avoiding trivial directions. We show that through a simple lightweight learner (SML), singular values provide compact information for effective merging coefficient predictions.
>
> We appreciate reviewer's comments and will append the rationale in the camera-ready version.
> ___
> R3W2 & R3Q3: *Theoretical analysis.*
>
> __AR3W2 & AR3Q3__: We acknowledge the reviewer's request of theoretical analysis. Our design intuitions, including statistical features with singular values from SVD and knowledge distillation for heterogeneous architectural merging, are mainly inspired by insights from __prior merging works__ (see Related Work section) and __extensive empirical results__ (at least 8 vision tasks and 6 NLP tasks). In this line of model merging research, e.g. Weight Averaging (Wortsman et al., 2022), Task Arithmetic (Ilharco et al., 2023), Fisher Merging (Matena and Raffel, 2022), RegMean (Jin et al., 2023), Ties-Merging (Yadav et al., 2023a) and AdaMerging (Yang et al.) and [A-C] from R2, the focus is primarily on empirical experiments without rigid formal proof in theory. In line with this direction, we have chosen to focus on extensive empirical evaluations and have not made any theoretical claims. Therefore, a rigorous theoretical analysis is beyond the scope of this work.
>
> However, we agree with the reviewer that theoretical analysis could make a valuable contribution to this field and represents a promising direction for future work. We will mention that in the Conclusion section in the camera-ready version.
>
>
> ___
> R3W3 & R3Q2: *Relationship between the proposed method and existing approaches, particularly AdaMerging.*
>
> __AR3W3__ & __AR3Q2__: We kindly refer the reviewer to Table 1: Summary of system characteristics in recent works in the __original submission__ which outlines the commonalities and differences between our proposed method and existing approaches, including AdaMerging.
>
> Extended explanation of the __Relationship with AdaMerging__:
> Both SML and AdaMerging share the spirit of employing a learner to predict merging coefficients. However, there are several key fundamental differences: (1) AdaMerging assumes architecturally identical whereas our distillation method __pioneers merging heterogeneous architectures__; (2) AdaMerging takes in the full model weights as input, whereas our proposed SML relies on __only the statistics of the weight distributions__, making it orders of magnitude more __lightweight__ (See __AR3W4__ & __AR3Q4__); (3) AdaMerging is trained by entropy minimization that encourages overconfident predictions that enforce the model towards spurious certainty especially on ambiguous or out-of-distribution data. In contrast, SML is trained on __pseudo labels produced by task-specific teachers__. Instead of reinforcing low-entropy predictions, SML distills knowledge from a trusted source, minimizing the risk of overconfidence in wrong directions using AdaMerging.
> ___
> R3W4 & R3Q4: *Experimental analysis on efficiency.*
>
> __AR3W4__ & __AR3Q4__: Our 2-Layer __StatsMergeLearner__ with the merged model contain 10.99M parameters, requires 2.95 GFLOPs, and achieves an inference time of 5.26 ms on an NVIDIA RTX A6000 GPU. We kindly refer to Table 4 in Sec. B.4 Efficient Inference in the Appendix in the __original submission__.
>
> Without the merged model, *StatsMergeLearner (SML)* itself is orders of magnitude __smaller__ and computationally __lighter__ than the merged model, with only 0.336M parameters, 0.73M MACs and 1.46M FLOPs. The results demonstrate that SML introduces __negligible overhead__ in terms of parameters (SML-to-Merged Model __Ratio__: 0.336M / 10.99M = __0.0306__) and computation (SML-to-Merged Model __Ratio__: 1.46M / 2.95G = __0.0005__).
>
> Lastly, we sincerely thank the reviewer for their time and effort in providing invaluable feedback. __ALL__ of the comments, including the related works mentioned by the reviewer, will be incorporated and reflected in the __camera-ready__ version. We welcome and encourage questions focused on the key contributions we have claimed: exploiting singular values from __SVD__, the __first heterogeneous architectural merging__ via distillation __without human-annotated labels__, and extensive empirical validation.

---

> > ### Comment · Reviewer_rPn2 · 2025-08-03
> > **New Official Comment**
> >
> > Thank you for your response. I have one follow-up concern regarding the authors' use of a simple lightweight learner operating on pseudo labels (in contrast to AdaMerging's approach in entropy space), which appears less tolerant to label noise. Specifically, I'd like to understand: the noise tolerance boundary where performance degrades significantly; the critical noise level causing model collapse; and the noise ratio range where this SML method demonstrates clear advantages over AdaMerging, for which empirical results or theoretical analysis would be greatly appreciated.

---

> > > ### Author Response · Authors · 2025-08-06
> > >
> > > Dear Reviewer rPn2 (R3),
> > >
> > > We hope our responses have addressed your concerns. Please let us know if you have any further questions or need additional clarification. Thank you!

---

> ### Author Response · Authors · 2025-08-03
> **Noise tolerance**
>
> Thank you for the thoughtful discussion regarding the tolerance to label noise between (1) AdaMerging's entropy-based and (2) our SML pseudo label-based optimization approach. We found that this question may offer deeper insights into the differing effects of (1) entropy and (2) pseudo labels.
>
> To design such experiments for empirical results, let's first quantify __noise level__ in this context. We use the __entropy__ of a task expert’s prediction (a model fine-tuned on task k) based on its output probability distribution. By using the __entropy__ as a proxy for the __label noise level__ of each sample, we partition the dataset and then compare the performance of AdaMerging and SML across different noise levels. Please let us know if this interpretation differs from your intent and we are happy to adjust the experiment settings. We are currently running these experiments and will share the results and insights shortly.

---

> ### Author Response · Authors · 2025-08-05
> **Noise tolerance boundary [1/3]**
>
> F: Follow-Up. A: Answer.
>
> We would like to thank Reviewer rPn2 (R3) again for the valuable question:
>
> R3F1: *noise tolerance boundary where performance degrades significantly; the critical noise level causing model collapse; and the noise ratio range where this SML method demonstrates clear advantages over AdaMerging.*
>
> __AR3F1.1 (label noise tolerance)__: Following the entropy metric described in the previous comment, we further normalized the entropy values to the range [0, 1] and split the dataset according to three noise levels evenly based on the normalized entropy: __Low noise__ [0, 0.33), __Medium noise__ [0.33, 0.66) and __High Noise__ [0.66, 1], where numbers represent the boundaries of the normalized entropy. We note that entropy computed in this way represents the confidence of a model, particularly the Task-Specific Teacher, acting as a proxy for label noise level, e.g. lower entropy indicates higher model confidence, and thus lower label noise, and vice versa. This interpretation aligns with its usage in the literature.
>
> While we are still waiting for confirmation from Reviewer rPn2 (R3) on whether this usage aligns with their intended meaning, we conducted experiments on eight vision tasks under three noise levels.
>
> Results are shown in Table AR3F1.1. We summarize the __new key insights__ as follows: both methods achieved their best performance on low-noise labels (as expected) with over 95% Avg Acc, and gradually degraded to around 80% as the noise level increased. Our proposed method, *StatsMerging*, consistently outperformed *AdaMerging++*, with performance gain of +4.4%, +6.2% and +8.5% under low, medium and high noise levels, respectively. Both methods appear to be learnable across three noise levels. We did not observe any noise level boundary where StatsMerging underperforms AdaMerging++.
>
> __Table AR3F1.1__. Comparison of StatsMerging and AdaMerging on Eight Vision Tasks under Three Noise Levels (Low, Medium and High). Numbers represent Avg Acc (%) across eight tasks.
>
> | Method             | Low | Medium | High |
> |--------------|-------|-------|-------|
> | AdaDerging++   | 95.5 | 91.4 | 80.1  |
> | __StatsMerging (Ours)__ | __99.9 (+4.4)__ | __97.6 (+6.2)__ | __88.6 (+8.5)__  |

---

> ### Author Response · Authors · 2025-08-06
> **Noise tolerance boundary [2/3]**
>
> __AR3F1.2 (input corruption tolerance)__: We would like to remind the reviewer that we also investigated the effect of input noise on images in Table 2 in Sec. B.2 Robustness Evaluation of our __initial submission__. This offers a different perspective to the label noise presented in AR3F1.1.
>
> For your convenience, we have included Table 2 below. In summary, *StatsMerging* outperforms AdaMerging under various types of image corruption, including Motion Blur, Impulse Noise and Gaussian Noise, demonstrating *StatsMerging*'s robustness.
>
> __Table 2__: Robustness results when merging ViT-B/32 models on four tasks. *StatsMerging*: shaded in gray. **Bold**: top score. Values are reported in %.
>
> | **Method**         | CA   | EU   | RE   | GT   | **Avg Acc**       |
> |--------------------|------|------|------|------|-------------------|
> | **Clean Test Set**                   |      |      |      |      | |
> | Task Arithmetic    | 66.9 | 94.7 | 82.6 | 75.1 | 79.8              |
> | AdaMerging         | 73.7 | 96.1 | 85.8 | 96.3 | 88.0              |
> | __StatsMerging__     | **75.6** | **96.3** | **92.1** | **97.6** | **90.4 (+2.4)** |
> | **Motion Blur**                   |      |      |      |      |    |
> | Task Arithmetic    | 65.3 | 68.1 | 80.0 | 64.2 | 69.4              |
> | AdaMerging         | 71.2 | 74.6 | 82.7 | 94.1 | 80.6              |
> | __StatsMerging__     | **73.5** | **76.9** | **89.2** | **95.2** | **83.7 (+3.1)** |
> | **Impulse Noise**                   |      |      |      |      |  |
> | Task Arithmetic    | 62.1 | 49.1 | 72.7 | 40.4 | 56.1              |
> | AdaMerging         | 67.2 | 30.8 | 75.9 | 77.5 | 62.8              |
> | __StatsMerging__     | **70.4** | **50.4** | **77.6** | **78.1** | **69.1 (+6.3)** |
> | **Gaussian Noise**                   |      |      |      |      | |
> | Task Arithmetic    | 63.6 | 55.4 | 75.9 | 49.4 | 61.1              |
> | AdaMerging         | 69.9 | 41.2 | 80.6 | 76.0 | 66.9              |
> | __StatsMerging__     | **71.2** | **53.6** | **82.1** | **78.0** | **71.2 (+4.3)** |

---

> ### Author Response · Authors · 2025-08-06
> **Noise tolerance boundary [3/3]**
>
> __AR3F1.3 (input noise tolerance)__: We additionally conducted experiments on merging two vision tasks on RESISC45 (RE) and EuroSAT (EU) on images with three levels of Gaussian noise: __Low noise__ ($\sigma=10$), __Medium noise__ ($\sigma=15$) and __High Noise__ ($\sigma=20$). Results are shown in Table AR3F1.3. In summary, as the noise level increased, the performance of both methods degraded. However, our proposed method *StatsMerging* consistently achieved higher accuracy than AdaMerging++ across all levels of Gaussian noise.
>
> __Table AR3F1.3__. Comparison of StatsMerging and AdaMerging on Two Vision Tasks RESISC45 (RE) and EuroSAT (EU) under Three Gaussian Noise Levels (Low, Medium and High). Numbers represent Avg Acc (%) across two tasks.
>
> | Method             | Low | Medium | High |
> |--------------|-------|-------|-------|
> | AdaMerging++   | 56.0 | 48.4 | 35.9  |
> | __StatsMerging (Ours)__ | __57.3 (+1.3)__ | __50.1(+1.7)__ | __36.7 (+0.8)__  |

---

### Official Review · Reviewer_zCNd · 2025-07-02

**Clarity:** 3
**Significance:** 2
**Originality:** 2
**Rating:** 3
**Confidence:** 4

**Summary:**

The paper introduces a new learning-based model merging method. The core idea is to leverage weight distribution statistics (singular values, mean, variances), which a lightweight learner, named StatsMergeLearner, takes as information for task improtance and predict task coefficients. Furthermore, the proposed method employs knowledge distillation to distill the knowledge from task-specific models into a merged model, when training StatsMergeLearner.

**Questions:**

Please refer to the weakness section.

**Ethical Concerns:**

["NO or VERY MINOR ethics concerns only"]

**Final Justification:**

While the rebuttal has made some clarifications, due to the concerns regarding the NLP experiments, large-scale vision experiments, and one of claimed contributions (knowledge distillation) as mentioned in the discussion, I will keep the original rating.

**Limitations:**

Yes.

**Quality:**

3

**Strengths And Weaknesses:**

Strengths:
- The paper is clear and easy to follow.
- The use of weight statistics to guide the prediction of task coefficients is straightforward and shown to be effective.


Weaknesses:
- The paper lack extensive experiments on larger scales and different domains. Particularly, in contrast to previous works (e.g., TALL-masks[A]), the paper does not show experimental results on ViT-L/14, 14 and 20 tasks, and also NLP tasks. The lack of such experiments raises concerns regarding the applicability and scalability of the proposed method.
- How does the performance differ when incorporating the distillation into AdaMerging, WEMoE [B], and TWIN-Merging [C]?
- The paper claims that the key benefit of employing distillation is reducing the reliance on human-annotated labels, however the paper performs distillation on data points (x_i) that already have labels.


Minor typos/grammar errors:
line 55: has far less explored -> has far less been explored
line 65: by leverage -> by leveraging
line 63 and Equation (1): W_k? or W_{\theta_k}?
line 193 is the repetition of  lines 184-192


[A] Wang et al., Localizing Task Information for Improved Model Merging and Compression. ICML 2024
[B] Shen et al., Merging multi-task models via weight-ensembling mixture of experts. ICML 2024
[C] Lu et al., Twin-Merging: Dynamic Integration of Modular Expertise in Model Merging. NeurIPS 2024

---

> ### Author Rebuttal · Authors · 2025-07-29
>
> We thank Reviewer __zCNd__ (__R2__) for acknowledging the __clear logic flow__ of our writing and for appreciating the __simplicity of using weight statistics to guide model merging__. The following responses address each concern individually, with __AR2W#__ denoting weaknesses and __AR2Q#__ denoting questions. A: Answer. T: Typo.
>
> ___
> R2W1.1: *Lack extensive experiments on larger scales.*
>
> __AR2W1.1__: We thank the reviewer for suggesting an extension of the experiments. Specifically based on this feedback, we conducted __additional__ experiments on __ViT-L/14__ across __eight diverse vision tasks__. Results in Table 1 shows that our proposed methods LW StatsMerging and LW StatsMerging++ achieve strong performance, __surpassing__ traditional multi-task learning baselines and prior model merging methods.
>
> These results confirm that our approach scales effectively to large models without any architectural modifications, demonstrating its practicality for large-scale model merging.
>
> __Table 1. Multi-task performance when merging ViT-L/14 models on eight tasks.__
> | **Method**                  | **SUN397** | **Cars** | **RESISC45** | **EuroSAT** | **SVHN** | **GTSRB** | **MNIST** | **DTD** | **Avg. Acc (%)** |
> |-|-|-|-|-|-|-|-|-|-|
> | Pre-trained                | 68.2       | 77.9     | 71.3         | 61.3        | 58.4     | 50.6      | 76.4      | 55.4    | 64.9     |
> | **Individual**             | **82.3**   | **92.4** | **97.4**     | **99.9**    | **98.1** | **99.2**   | **99.7**  | **84.1**| **94.1**  |
> | Traditional MTL            | 80.8       | 90.6     | 96.3         | 96.3        | 97.6     | 99.1      | 99.6      | 84.4    | 93.5     |
> | | | | | | | | | | |
> | Weight Averaging           | 72.1       | 81.6     | 82.6         | 91.4        | 78.2     | 70.6      | 97.0      | 62.8    | 79.5     |
> | Fisher Merging             | 69.2       | 88.6     | 87.5         | 95.5        | 80.6     | 74.8      | 93.3      | 70.0    | 82.2     |
> | RegMean                    | 73.3       | 81.8     | 86.1         | 92.4        | 82.8     | 84.2      | 98.5      | 60.8    | 82.5     |
> | Task Arithmetic            | 74.1       | 82.1     | 87.7         | 92.6        | 87.9     | 84.0      | 98.6      | 65.5    | 84.4     |
> | TIES-Merging               | 75.0       | 84.5     | 88.0         | 94.3        | 85.7     | 88.1      | 98.7      | 67.7    | 84.5     |
> | LW AdaMerging | 79.0       | 90.3     | 90.8         | 96.2        | 93.4     | 98.0      | 99.0      | 79.9    | 90.8     |
> | LW AdaMerging++ | 79.4       | 90.3     | 91.6         | 97.4        | 93.4     | 97.6      | 99.0      | 79.2    | 91.0     |
> | WEMoE                      | 81.4       | 92.6     | 95.4         | 99.4        | 97.7     | 99.9      | 99.7      | 83.7    | 93.6     |
> | **LW StatsMerging (Ours)** | 80.6       | 90.5     | 94.7         | 96.8        | 93.6     | 98.3      | 98.9      | 83.2    | 92.1     |
> | **LW StatsMerging++ (Ours)** | **82.2** | **92.8** | **97.2**     | **99.3**    | **97.9** | **99.5**   | **99.8**  | **84.2**| **94.1**  |
>
> R2W1.2: *Experiments on NLP tasks.*
>
> __AR2W1.2__: Please refer to __AR1W3.2: Additional results in NLP tasks.__ In summary, our method achieves an Avg Acc of 77.6%, 3.7% higher than the second best method.
> ___
> R2W2: *Effect of Incorporating Distillation into AdaMerging, WEMoE, and TWIN-Merging.*
>
> __AR2W2__: We appreciate the reviewer’s suggestion. To analyze the benefit of our distillation framework, we incorporated the distillation step into AdaMerging and WEMoE. As shown in Table 2, incorporating distillation improves the performance of these merging methods on CIFAR-100 (CI), Cars (CA), and EuroSAT (EU) tasks. The largest gains are observed for AdaMerging, while LW StatsMerging achieves the highest average accuracy overall. Distillation is performed from a teacher ViT-B/16 (VI) to a ResNet-50 (RN) student before merging. Top-1 accuracy (%) is reported on CIFAR-100 (CI), Cars (CA), and EuroSAT (EU) for various merging methods.
>
> __Table 2. Knowledge distillation improves model merging performance__
> | **Method**                     | **CI**   | **CA**   | **EU**   | **Avg Acc (%)** |
> |-------------------------------|----------|----------|----------|-------------------|
> | Backbone Distilled            | RN       | VI       | VI       | –                 |
> |                               | –        | RN     | RN     | –                 |
> | Individual Distilled          | 97.8     | 77.7     | 99.7     | 91.7              |
> |                               | –        | 76.4     | 94.5     | –                 |
> | Multitask Distilled | 96.4| 74.6 | 96.2 | 89.1 |
> |                               |          |          |          |                   |
> | Weight Averaging              | 77.1     | 56.4     | 64.9     | 59.4              |
> | Task Arithmetic               | 76.5     | 52.8     | 80.1     | 69.8              |
> | TIES-Merging                  | 81.4     | 61.6     | 78.2     | 73.7              |
> | AdaMerging     | 84.9     | 65.1     | 85.7     | 78.6              |
> | WEMoE          | 86.5     | 67.2     | 87.6     | 80.4              |
> | **LW StatsMerging (Ours)**    | **87.2** | **68.4** | **88.4** | **81.3**          |
> ___
> R2W3: *Clarity of distillation without human-annotated labels.*
>
> __AR2W3__: We appreciate the reviewer’s observation. The key advantage of our distillation-based approach is that, during the merging phase, the method does __not__ require __human-annotated labels__ for supervision. Instead, the teacher models provide __soft targets (pseudo-labels)__ that guide the training of the merged model.
>
> Although our experiments were conducted on datasets that contain human-annotated labels, these labels are __never__ in the __distillation__ phase. The teacher models generate predictions, and the merged model is trained to match these outputs. This setup can be applied equally well to unlabeled data or out-of-distribution samples, as the teachers can provide pseudo-labels without human annotation.
>
> We will also include additional results in the camera-ready version to explicitly demonstrate merging on unlabeled data using teacher-provided pseudo-labels.
> ___
> __AR2T__: We thank Reviewer zCNd for the suggestions of correcting the minor issues of typos. We will incorporate the corrections in the camera-ready version.
>
> Lastly, we sincerely thank the reviewer for their time and effort in providing invaluable feedback. __ALL__ of the comments, including the related works mentioned by the reviewer, will be incorporated and reflected in the __camera-ready__ version. We welcome and encourage questions focused on the key contributions we have claimed: exploiting singular values from __SVD__, the __first heterogeneous architectural merging__ via distillation __without human-annotated labels__, and extensive empirical validation.

---

> > ### Author Response · Authors · 2025-08-03
> >
> > Dear Reviewer zCNd (R2), we sincerely appreciate the time and effort you invested in your detailed reviews. We hope our responses have addressed your concerns, and we would be happy to provide further clarification if needed. Thank you again for your thoughtful feedback.

---

> > > ### Comment · Reviewer_zCNd · 2025-08-05
> > >
> > > Thank you for the rebuttal.
> > >
> > > However, the rebuttal does not address all the concerns raised in the original review.
> > > Many papers show the performance on 14 and 20 vision tasks as stated in the review. However, the rebuttal does not show the results on this setting, as done in [A] referred in the original review.
> > >
> > > The models compared in NLP tasks seem a bit outdated; TIES-Merging was published two years ago. To the support the claim that the proposed method demonstrates the outstanding the performance, the comparisons against AdaMerging, TALL-Mask [A], WEMoE [B], and Twin-Merging [C].
> > >
> > > The experimental settings on the distillation still remain unconvincing, as the available labels are forcibly unused.
> > > The paper could have included the results on actual unlabeled datasets to support the claim.
> > > Furthermore, the use of distillation is not a novel idea and well known to have positive effects on the performance.

---

> ### Author Response · Authors · 2025-08-06
>
> Thank you for your continued feedback. We respectfully disagree with some of the points raised.
>
> Regarding the use of distillation, our intention is not to claim novelty in the distillation method, but rather in how it is applied in the model merging context. While knowledge distillation is a widely used technique in supervised learning, to the best of our knowledge, it has not been explored in the setting of model merging, particularly in the absence of labelled data. Most existing approaches rely on labelled validation sets for merging or post-hoc adaptation. In contrast, our method uses teacher models as the source of supervision and can effectively merge models without access to any ground-truth labels, which is a unique and practical advantage.
>
> __AR2 The experimental settings on the distillation:__ We would like to emphasise that the decision to exclude labelled validation data was intentional, not a constraint. Our goal is to demonstrate the ability of our method to operate in scenarios where task labels are unavailable, which is a common challenge in practice. In this way, our distillation framework provides a compelling and novel solution that goes beyond conventional settings by enabling task-specific model merging using only the original models themselves. This capability distinguishes our approach from prior works that require labelled data to achieve strong performance.
>
>
> __NLP Tasks :__ We appreciate the reviewer’s comments regarding the evaluation setup and selection of baselines. Due to computational constraints, we prioritized task diversity across both vision and NLP domains rather than an exhaustive evaluation on 14 or 20 vision tasks as done in [A]. Nonetheless, we agree that such extended vision-only evaluations are valuable and will consider them in future work. On the NLP side, while TIES-Merging is indeed two years old, we included it for completeness alongside more recent and competitive baselines such as LiNeS and KnOTS. We acknowledge the importance of comparisons against additional methods like AdaMerging, TALL-Mask, WEMoE, and Twin-Merging. However, we note that AdaMerging relies on entropy-based confidence, which is more naturally suited to vision tasks, and was originally evaluated in that context. In contrast, our inclusion of NLP tasks is specifically to demonstrate task diversity and the generalizability of our approach. We will aim to incorporate these additional baselines in future work, where compatible implementations and checkpoints are publicly available.

---

> > ### Comment · Reviewer_zCNd · 2025-08-06
> >
> > Thank you for the response.
> >
> > Regarding the the importance of distillation, I understand the motivations and intent of authors. However, the distillation does not appear as one of major technical contributions/key advantages, in contrast to the claim in the abstract and throughout the paper.
> >
> > The inclusion of recent competitive models (AdaMerging, WEMoE) in vision experiments but not in NLP experiments makes NLP experiments less convincing.
> >
> > I will keep the rating for now, and discuss with other reviewers to hear their opinions to reach the final conclusion.

---

> > > ### Author Response · Authors · 2025-08-09
> > >
> > > Both AdaMerging WEMoE are evaluated on vision tasks only in their reports. We are committed to include recent advanced methods in the NLP tasks in the camera-ready version. Please refer to __AR1W3F1__.

---

> ### Author Response · Authors · 2025-08-08
> **R2F1: Distillation major technical contributions/key advantages**
>
> Thank you for providing the follow-up questions.
>
> F: Follow-Up. A: Answer:
>
> R2F1: "the distillation does not appear as one of major technical contributions/key advantages, in contrast to the claim in the abstract and throughout the paper."
>
> __AR2F1__: We __pioneer__ the use of distillation in model merging research in two ways, as outlined in the Abstract. We demonstrate the empirical advantages across eight vision and six NLP tasks, and show clear improvements over the Multitask distillation baseline (__UAR1W2__), as well as more resilient noise tolerance than AdaMerging (__AR3F1.1__).

---

### Official Review · Reviewer_yiJc · 2025-07-03

**Clarity:** 2
**Significance:** 2
**Originality:** 2
**Rating:** 4
**Confidence:** 5

**Summary:**

The paper titled “StatsMerging: Statistics-Guided Model Merging via Task-Specific Teacher Distillation” introduces StatsMerging, a lightweight, learning-based method for predicting task/layer-specific scaling coefficients for model merging. The approach utilizes statistical features—such as the mean, variance, magnitude, and singular values of model weights—which are input to a lightweight two-layer MLP. This MLP is jointly trained across all tasks\layers to predict merging coefficients for each task/layer. Additionally, the method employs knowledge distillation to align the architectures of task-specific experts, enabling a distill-then-merge paradigm.

**Questions:**

- While the paper demonstrates strong results using a simple 2-layer MLP, it remains unclear what the optimal network size is. Would increasing the model capacity—e.g., using a slightly larger network—lead to further improvements?

- The paper lacks clarity regarding the construction of matrix of model parameters used to compute the singular values across the entire model. Further, Lines 163-164 mention the use of rank-3 from \Sigma_k, however it is unclear why this choice was made. Can the authors clarify this?

- While Task Arithmetic and TIES are widely used merging techniques, recent methods such as LiNeS [1] and KnOTS[2] have demonstrated substantial improvements over these baselines. Incorporating these state-of-the-art methods—both as baselines and in combination with SML—would provide a clearer picture of the performance ceiling and better contextualise the gains achieved by the proposed approach.

- Lines 184-192 repeat themselves at line 193, the authors are advised to remove this redundancy.

References:
- [1] Wang, Ke, et al. "Lines: Post-training layer scaling prevents forgetting and enhances model merging." arXiv preprint arXiv:2410.17146 (2024).
- [2] Stoica, George, et al. "Model merging with SVD to tie the Knots." arXiv preprint arXiv:2410.19735 (2024).

**Ethical Concerns:**

["NO or VERY MINOR ethics concerns only"]

**Final Justification:**

I would like to thank the authors for the clarification. While some concerns (W1, W3, W4(part-1)) still remain, I'm happy with the additional evaluation on comparing distill+merge with multitask-distillation and am willing to raise my score.

**Limitations:**

yes

**Quality:**

2

**Strengths And Weaknesses:**

### Strengths:
- The idea of using statistical features as inputs to a lightweight network trained using distilled labels to predict scaling coefficients for model merging is simple and novel.

- The paper introduces a distil-then-merge paradigm, leveraging knowledge distillation to convert task-specific experts with heterogeneous architectures into ones with the same architecture, enabling model merging—a direction that has not been explored previously.

### Weaknesses:
- It is unclear whether the StatsMergeLearner (SML) network, once trained on a set of tasks, could potentially generalise to unseen tasks or architectures. Demonstrating such generalisation would significantly enhance the utility and applicability of the method.

- Although the paper introduces the distill-then-merge paradigm, it lacks a comparison with a multitask distillation baseline, which could offer a more cost-effective alternative with potentially better performance.

- Evaluation is limited to vision tasks, overlooking the language domain where model merging techniques are more prevalent and extensively studied.

- The method has only been tested on relatively small architectures (<100M parameters), such as ViT-B/32 and ResNet-50. In order to highlight the applicability of the work at scale, the authors are advised to test on much larger models.

---

> ### Author Rebuttal · Authors · 2025-07-28
>
> We thank Reviewer __yiJc__ (R1) for acknowledging the __simplicity__ and __novelty__ of our proposed method, as well as recognizing our work as a __pioneering__ effort in a new direction by introducing the __distill-then-merge__ paradigm as the __first__ approach to merging __heterogeneous architectures__.
>
> We also appreciate Reviewer __yiJc__'s time and effort in providing feedback to improve our work, particularly regarding clarity, detail, and comparison with recent works. We address these concerns one by one in the following responses noted as __AR1W#__ for weaknesses and __AR1Q#__ for questions. A: Answer.
>
> ___
> R1W1: *Generalising to unseen tasks or architectures?*
>
> __AR1W1__: __Yes__.
>
> __AR1W1.1__: __Unseen tasks__. We thank the reviewer for the valuable suggestion regarding generalization evaluation. As noted, we have conducted corresponding experiments and included the results in the __original submission__. We kindly refer to the details from in Line 238 Significantly Enhanced Generalization and Table 3: Generalization results (Avg Acc %) on two unseen tasks when merging Layer-Wise ViT-B/32 models on six tasks. We quote the summarized results to address this question: Our proposed *StatsMerging* achieved 70.80% and 60.70% Avg Acc, outperforming the state-of-the-art AdaMerging++ which achieved 68.70% and 58.50%, respectively.
>
> __AR1W1.2__: __Heterogeneous architectures (arch)__. We first thank the reviewer for __acknowledging__ that both ViT-B/32 and ResNet-50 (__different arch__) were evaluated in __R1W4__. Please also refer to the details in Sec. Extension to Heterogeneous Architectures from Line 250 and Table 3, 5 in the __original submission__. In summary, we demonstrated the extension to __ResNet50__ while the primary arch is __ViT__, achieving 81.30% Avg Acc compared to Ties-Merging at 69.80% and Task Arithmetic at 73.7%, respectively.
> ___
> R1W2: *Multitask distillation baseline*.
>
> __AR1W2__: Please refer to the results in *Table 2. Knowledge distillation improves model merging performance* in __AR2W2__ (Row Name: *Multitask Distilled*).
>
> In summary, the multitask distillation baseline (Row Name: *Multitask Distilled*) achieves 89.1% average accuracy, slightly below the 91.7% upper bound achieved by the individual distillation baseline (Row Name: *Individual Distilled*). Note that this multitask distillation baseline follows conventional multitask learning, which requires human annotations to update the entire set of shared weights in each epoch. In contrast, our proposed method __LW StatsMerging__ does not use any human-annotated data. It only updates the lightweight SML weights (3.06% SML-to-Merged Model Ratio shown in __AR3W4__ & __AR3Q4__) in each epoch, and achieves the __best performance__ at 81.3% among the model merging methods ranging from the Weight Averaging baseline to WEMoE.
> ___
> R1W3: *Evaluation is limited to vision tasks, overlooking the language domain.*
>
> __AR1W3__: We thank the review for the insightful suggestion of extending to the language domain. We address this question in three perspectives:
>
> __AR1W3.1__: __Vision is the main focus of this work__. As stated on Line 55, adapting these merging techniques in computer vision domain has far less explored in the literature. We also acknowledged extending this task to language tasks as future work in on Line 127 in Section B.7 (Future Work and Limitations) in the Appendix.
>
> __AR1W3.2__: __Consistent vision evaluation protocol with prior works__. We followed the evaluation protocol in the prior work AdaMerging (Line 241), which also purely focused on vision tasks __without__ NLP tasks.
>
> __AR1W3.2__: __Additional results in NLP tasks__.  To demonstrate the applicability of our method to the language domain, apart from the experiments in the original submission, in this rebuttal we further conducted additional experiments on __six NLP__ benchmarks (paws, qasc, quartz, story cloze, wiki qa, winogrande, wsc) using *LW StatsMerging*. As shown in __Table 1__, our method achieves an Avg Acc of 77.6%, 3.7% __higher than the second best__ method. Results confirm that our framework can be directly extended to language tasks. We plan to incorporate large language-specific adaptations in future work to further improve performance. We will release the code in the provided github URL.
>
>
> __Table1. Evaluation of model merging methods on six NLP tasks__
> |**Method**|**Val**|**paws**|**qasc**|**quartz**|**story_cloze**|**wiki_qa**|**winogrande**|**wsc**|**Avg Acc**|
> |-|-|-|-|-|-|-|-|-|-|
> |Zeroshot|–|49.9| 35.8| 53.3| 48.1| 76.2| 50.0| 61.1|53.5|
> |Fine-tuned|–|94.3|98.3|80.4|84.7|95.5|64.1|62.5|82.8|
> |Multitask|–|94.0|97.9|82.5|86.7|95.0|64.1|65.3|83.6|
> |||||||||||
> |Weight Averaging|✗|66.4|82.6|60.2|49.5|94.1|50.4|58.3|65.9|
> |Task Arithmetic|✗|73.3|93.5|68.2|76.5|93.7|55.5|56.9|73.9|
> |TIES-Merging|✗|74.0|83.3|70.3|64.2|84.7|55.9|55.6|69.7|
> |Fisher Merging|✓|69.3|85.7|63.6|56.4|93.8|50.9|62.5|68.9|
> |RegMean|✓|76.8|96.2|62.5|55.0|94.8|51.9|61.1|71.2|
> |Task Arithmetic|✓|73.4|94.3|67.1|71.7|94.1|52.9|59.7|73.2|
> |TIES-Merging|✓|79.3|88.6|71.8|72.9|82.5|61.3|61.1|73.9|
> |**LW StatsMerging (Ours)**|✓|**82.1**|**96.2**|**73.2**|**73.1**|**94.9**|**62.1**|**62.2**|**77.6 (+3.7)**|
> ___
> R1W4: *Test on much larger models.*
>
> __AR1W4__: We thank the reviewer for this valuable suggestion. To study the effect of increasing the model capacity of StatsMergeLearner (SML), we performed experiments on two datasets (RESISC45, EuroSAT) under three settings:
>
> 1) the current 2-layer MLP,
> 2) an MLP with 4 layers,
> 3) replacing the MLP with a lightweight Transformer.
>
> As shown in Table 2, increasing capacity leads to slightly better accuracy and faster convergence. Notably, replacing the MLP with a Transformer yields the best performance and fastest convergence. However, such performance gains are marginal such as only a +0.1 or +0.3 increase, and come at the cost of doubling the number of parameters using a 4-layer MLP or a more complex Transformer architecture. The results indicate that using a 2-Layer MLP offers a __good trade-off__ between performance and model complexity. We intentionally retain the __simplicity__ and __lightweight__ nature of the neural network design in the proposed StatsMergeLearner (SML), using only 2 layers to highlight the effectiveness of the overall framework shown in Fig. 2.
>
> __Table 2. Avg Acc (%) Performance of SML with different capacities on RESISC45 and EuroSAT__ *: Current capacity in the submission.
>
> |**SML Design Choice**|**RESISC45**|**EuroSAT**|**Avg Acc (Epoch)**|**#Params (M)**|MACs (M)|FLOPs (M)|
> |-|-|-|-|-|-|-|
> |Individual Model|96.1|99.7|97.9|-|-|-|
> |||||
> |2-Layer MLP*|96.0|98.0|97.4 (Epoch 750)|0.366|0.73|1.46|
> |4-Layer MLP|97.0|98.0|97.5 (+0.1, Epoch 610)|0.732|1.46|2.91|
> |2-Layer Transformer|97.0|98.5|97.7 (+0.3, Epoch 426)|0.396|0.79|1.58|
>
> For the camera-ready draft, we will further include results on merging tasks $>2$.
>
> ___
> R1Q1: *Effect of model size on performance. A larger network can lead to further improvements?*
>
> __AR1Q1__: Yes. Please refer to __AW4__.
> ___
> R1Q2.1: *Clarity regarding the construction of matrix for SVD.*
>
> __AR1Q2.1__: We apologize for the lack of clarity. The construction of the parameter matrix for singular value decomposition (SVD) is as follows: For each layer k, we flatten its parameter tensor Wk into a 2D matrix. For linear layers, this is typically (out features, in features). For convolutional layers with a kernel of shape (out channels, in channels, kernel height, kernel width), we reshape it to (out channels, in channels × kernel height × kernel width). We then compute the SVD:
>
> $W_{K} = U_{k} \Sigma_{k} V^{T}_{k}$
>
> and extract the singular values from $\Sigma_k$. The singular values across all layers are concatenated to form
> the feature vector used as input to SML.
>
> We will include these details in Sec. 3.2 Weight Statistics-Guided Model Merging in the camera-ready version.
> ___
> R1Q2.2: *Justification of Rank-3 design choice.*
>
> __AR1Q2.2__: We selected singular values based on an empirical observation that the top-3 singular values capture much of the the spectral energy in our experiments. Using higher ranks leads to negligible improvements while increasing computation.
> ___
> R1Q3: *Compare KnOTS and combine it with the proposed SML.*
>
> __AR1Q3__: We appreciate the reviewer’s suggestion. We included KnOTS as an additional baseline and evaluated StatsMerging + KnOTS. As shown in Table 3, StatsMerging + KnOTS performs worse than our proposed StatsMerging in this two-task setting. We hypothesize that this is due to (i) KnOTS being sensitive to SVD rank selection and scaling, and (ii) its design being more beneficial for larger and more diverse task sets. Although KnOTS converges faster, it incurs approximately __10× higher training cost per epoch__ due to repeated SVD computations. To further investigate KnOTS, we are currently running experiments on 8 datasets, and we will include these insights in the camera-ready version.
>
> __Table 3. Comparison of different merging methods on two-task merging (RESISC45, EuroSAT)__
> |**Method**|**RESISC45**|**EuroSAT**|**Avg Acc (%)**|
> |-|-|-|-|
> |Individual|96.1|99.7|97.9|
> |Task Arithmetic|85.2|96.7|90.9|
> |TIES-Merging|86.4|97.2|91.8|
> |**StatsMerging + KnOTS**|**92.1**|**94.2**|**93.2**|
> |StatsMerging (Ours)|96.0|98.0|97.4 (+4.2)|
> ___
> __AR1Q4__: We thank the reviewer for the feedback and will remove the redundant paragraph in the camera-ready version.
>
> Lastly, we sincerely thank the reviewer for their time and effort in providing invaluable feedback. __ALL__ of the comments, including the related works mentioned by the reviewer, will be incorporated and reflected in the __camera-ready__ version. We welcome and encourage questions focused on the key contributions we have claimed: exploiting singular values from __SVD__, the __first heterogeneous architectural merging__ via distillation __without human-annotated labels__, and extensive empirical validation.

---

> > ### Comment · Reviewer_yiJc · 2025-08-06
> > **Further Clarifications**
> >
> > I’d like to thank the authors for their response. Below are some follow-up comments and clarifications:
> >
> > **W1:** My intent was to assess the generalization ability of the SML network rather than the merged model itself. Specifically, the goal is to evaluate whether a trained SML network can be reused to merge an unseen set of expert models (potentially with different architectures) than those used during its training. For example, within the 8-task vision benchmark, this could involve training the SML network on four vision experts (each trained on distinct, seen tasks), then using it to generate scaling coefficients for four unseen experts (trained on unseen tasks) and evaluating the resulting merged model (from the four unseen experts) on those unseen tasks.
> >
> > **W2:** Multitask distillation is not the same as multitask training, as the former differs by making use of pseudo-labels from multiple teacher models, mitigating the need for human annotation.
> >
> > The authors could confirm if the multitask distillation baseline, achieving 89.1% in AR2W2, was obtained using pseudo labels from the expert models. If so, this clearly showcases the superiority of the simpler multitask distillation approach over distill+merge.
> >
> > **W3:** I appreciate the inclusion of evaluations on the NLP benchmark. While it is encouraging to see LW Statsmerging outperform training-free merging methods, the absence of the key baseline AdaMerging/AdaMerging++ makes it difficult to fully contextualize its performance. Additionally, could the authors clarify the model architectures used for the expert models in this evaluation?
> >
> > **W4:** Though this is a secondary concern, my earlier point aimed to probe SML’s performance when merging large size expert models (1-8 billion parameter size), rather than increasing the capacity of the SML network itself. It is also common to report results across both ViT-B/32 and ViT-L/14 for analyzing performance gains with increasing expert model size.

---

> > > ### Author Response · Authors · 2025-08-07
> > > **Response to W2: Multitask distillation**
> > >
> > > Regarding __W2: Multitask distillation__, there appears to be a mismatch in the specific setting of this approach in our previous communication. We thank Reviewer yiJc (R1) for taking the time to provide further clarification in the discussion period. We have obtained some results and will provide an updated response along with some corrections by the end of today.

---

> ### Author Response · Authors · 2025-08-03
>
> Dear Reviewer yiJc (R1), we hope our responses address your concerns. If you have any further questions or require clarification, please let us know. Thank you!
>
> Meanwhile, I would like to make a minor correction in the main Rebuttal:
>
> ___
>
> [Original text]
>
> ->
>
> [Corrected text]
>
> ___
>
> __AR1Q1__: Yes. Please refer to __AW4__.
>
> ->
>
> __AR1Q1__: Yes. Please refer to __AR1W4__ & __AR2W1.1__.
>
> ___

---

> ### Author Response · Authors · 2025-08-06
> **Larger model of ViT-L/14 in W4**
>
> We thank Reviewer yiJc (R1) for their further clarification. We would have appreciated it if these clarifications had been provided earlier.
>
> ___
>
> A: Answer
>
> __AR1W4__: For W4 please refer to __ViT-L/14__'s performance in __AR2W1.1__ stated in the previous comment.

---

> ### Author Response · Authors · 2025-08-08
> **Updated Response to W2: Multitask distillation**
>
> W2: *Multitask distillation*.
>
> We sincerely apologize for the miscommunication and appreciate the opportunity to provide additional results and corrections based on the reviewer's further clarifications.
>
> U: Updated. A: Answer.
>
> __UAR1W2__:
>
> In the updated Table 2 (originally from __AR2W2__), we corrected Multitask Distilled (MTD) to Multitask Learning (MTL), with * indicating the change. MTL trains a single shared model on all tasks using human-annotated labels (GT). MTD is identical except it uses only pseudo labels (instead of GT), which are predictions from Task-Specific Experts for training.
>
> **Updated Table 2. Knowledge distillation improves model merging performance (%)**
>
> | Method                   | CI   | CA   | EU   | Avg Acc |
> |--------------------------|------|------|------|--------------|
> | Backbone Distilled       | RN   | VI   | VI   | –            |
> | –                        | RN   | RN   | –    |              |
> | Individual Distilled     | 97.8 | 77.7 | 99.7 | 91.7         |
> | –                        |      | 76.4 | 94.5 | –            |
> | \*Multitask Learning (MTL)\*      | 96.4 | 74.6 | 96.2 | 89.1         |
> | \***Multitask Distilled (MTD)**\*      | 89.3 | 52.7  | 83.4 | **75.1**         |
> | Weight Averaging         | 77.1 | 56.4 | 64.9 | 59.4         |
> | Task Arithmetic          | 76.5 | 52.8 | 80.1 | 69.8         |
> | TIES-Merging             | 81.4 | 61.6 | 78.2 | 73.7         |
> | AdaMerging               | 84.9 | 65.1 | 85.7 | 78.6         |
> | WEMoE                    | 86.5 | 67.2 | 87.6 | 80.4         |
> | **LW StatsMerging (Ours)** | 87.2 | 68.4 | 88.4 | **81.3**     |
>
> We summarize the key findings of Updated Table 2:
>
> **(1)** Our proposed **LW StatsMerging achieves the highest performance** of 81.3% Avg Acc, significantly outperforming the Multitask Distillation baseline at 75.1% shown in the row Multitask Distilled (MTD).
>
> **(2)** We hypothesize that **pseudo label quality can significant affect task interference**. Observe in the upper bound of Individual Distilled results, there exists a highly imbalanced performance between task CA (77.7%) and the other two tasks, CI (97.8%) and EU (99.7%). While MTL CA (74.6%) CI (96.4%) EU (96.2%) achieves near the Individual Distilled performance, it comes at the cost of expensive human-annotated ground truth.
>
> Although the overall MTD performance (75.1%) is better than the other two basic model merging baselines of Weight Averaging (59.4%) and Task Arithmetic (69.8%), for the the CA task, MTD's (52.7%) scores lower than both baselines of  56.4% and 52.8%, respectively. Since CA is a more difficult task (lower accuracy), the CA Task-Specific Teacher could produce less confident predictions with higher entropy. We therefore conjecture that low-quality pseudo labels from the CA Task-Specific Teacher may provide limited benefit to (or even worsen) the learning of the model distilled by MTD.
>
> **(3)** **Mechanisms that are originally designed to address the challenges of task inference in advanced model merging methods can also help mitigate the negative effects of low-quality pseudo labels**. We observe that all recent advanced model merging methods, including TIES-Merging, AdaMerging, WEMoE, and LW StatsMerging, substantially improve CA task performance, raising it from 52.7% (MTL) to at least 61.6% (TIES-Merging) with an 8.9% increase.
>
> We speculate that the improvement stems from task interference mitigation mechanisms in model merging techniques, such as Elect Sign in TIES-Merging, the router network for handling conflicts between task-specific parameters in WEMoE, entropy-minimization-based merging coefficient learning in AdaMerging, and SML in StatsMerging. Among these, our proposed distillation-based SML learns the weight coefficients to balance pseudo label quality, achieving the highest score of 68.4% in this CA task.
>
> We believe these new insights can help shape the next new design of future model merging methods in the knowledge distillation direction.
> ___
>
> F: Follow-Up.
>
> R1W2F1: *The authors could confirm if the multitask distillation baseline, achieving 89.1% in AR2W2, was obtained using pseudo labels from the expert models.*
>
> __AR1W2F1__: __No__. This refers to __multitask training/learning__, which corresponds to the row \*Multitask Learning (MTL)\* in Updated Table 2. We apologize for the miscommunication and thanks for the further clarification again. The Multitask distillation (MTD) results are in __UAR1W2__.
>
> The following text (originally in __AR1W2__) is corrected as follows with * highlighting the change:
>
> "*Note that this multitask distillation baseline follows conventional multitask learning, which requires human annotations to update the entire set of shared weights in each epoch.*"
>
> ->
>
> "*Note that this multitask \*training\* baseline follows conventional multitask learning, which requires human annotations to update the entire set of shared weights in each epoch.*"

---

> > ### Comment · Reviewer_yiJc · 2025-08-08
> >
> > I would like to thank the authors for the clarification. While some concerns (W1, W3, W4(part-1)) still remain, I'm happy with the additional evaluation on comparing distill+merge with multitask-distillation and am willing to raise my score.

---

> > > ### Author Response · Authors · 2025-08-08
> > > **Thank You**
> > >
> > > We thank Reviewer __yiJc__ (R1) for acknowledging our efforts and for the encouraging feedback. Our team is working to address the remaining concerns. If you have any other questions, please let us know. Thank you!

---

> ### Author Response · Authors · 2025-08-08
> **Follow-Up Response to R1Q3: Comparison with LiNeS**
>
> F: Follow-Up. A: Answer.
>
> R1Q3F1: *Compare LiNeS and combine it with the proposed SML.*
>
> __AR1Q3F1__:
>
> __Combined with SML?__
>
> *Similarity*:
>
> Both share a similar goal of preserving common features across tasks while retaining task-specific representations
>
> *Difference*:
>
> LiNeS scales the updates from shallow to deep layers linearly, controlled by $\alpha$ and $\beta$. In Layer-Wise (LW) StatsMerging, merging coefficients ($\lambda$) are optimized across the entire merged model by SML, therefore in theory $\lambda$ should jointly account for the scales of updates from shallow to deeper layers. In addition, SML does not assume the linear scaling from shallow to deeper layers as in LiNeS.
>
> We therefore posit that SML (and other learning-based methods) may not benefit significantly from directly integrating LiNeS scaling coefficients, either during training or in post-training stages. This is consistent with the fact that in the LiNeS paper, the merging methods that LiNeS integrates with are heuristic-based, including Task Arithmetic, Ties-Merging, Consensus Merging (Table TAR1Q3F1) and Model Soup. The only learning-based method reported in the experiments is AdaMerging, which was only used solely for comparison, if I am not mistaken. Although SML can be combined with LiNeS in practice/implementation, we find it theoretically unnecessary.
>
> __Comparison__
>
> We present the comparison of LiNeS and our updated StatsMerging (w SML) on merging ViT-B/32 in Table TAR1Q3F1. Our proposed StatsMerging (84.5%) significantly outperforms the best reported LiNeS result (77.2%).
>
> Table __TAR1Q3F1__: Multi-task merging performance (Avg Acc %) when merging ViT-B/32 models on eight tasks. Results of our method *StatsMerging* are in bold. **LW:** Layer-wise.
>
> | **Method**         | **Avg Acc**       |
> |--------------------|------|
> | Task Arithmetic | 69.7 |
> | Task Arithmetic + LiNeS | 74.2 |
> | Ties-Merging | 73.6 |
> | Ties-Merging + LiNeS | 77.2 |
> | Consensus Merging | 74.5 |
> | Consensus Merging + LiNeS | 77.6 |
> | LW AdaMerging | 80.1 |
> | LW AdaMerging++ | 81.1 |
> | **LW StatsMerging (Ours)** | **84.5** |
>
> We will summarize the results and discussion on LiNeS with/vs SML in the camera-ready version to inspire new weight-scaling methods for model merging in future work.

---

> ### Author Response · Authors · 2025-08-09
> **Response to W1: Generalization Ability of SML**
>
> F: Follow-Up. A: Answer.
>
> R1W1F1: *assess the generalization ability of the SML network rather than the merged model itself*.
>
> __AR1W1F1__: We use SML trained on the eight vision datasets (LW StatsMerging), where it was exposed to solely the ViT-B/32 architecture for Task-Specific Experts on each vision task. This SML is then used to generate merging coefficients for merging two __unseen__ ResNet50 models, each pre-trained on __unseen__ CIFAR10 (CF10) and CIFAR100 (CF100) tasks. We evaluate SML in the Layer-Wise (LW) setting. This setup is summarized in Table __TAR1Q3F1.1__.
>
> Table __TAR1Q3F1.1__: Generalization Experiment Setup
> | | Architecture Type | Architecture | Task |
> |---|---|---|---|
> | Train | Task-Expert | ViT-B/32 | SU CA RE EU SV GT MN DT |
> | Test | Merged | ResNet50 | CF10 CF100 |
>
> *Challenge*: Mismatch Layer. To generalize to a different architecture, we encountered the new challenge that the expert and the merged model layers differ. We subsample 22 coefficients to merge ResNet50 models from the 320 ViT-B/32 coefficients, enforcing consistency in the relative positions of coefficients and layers across both architectures. This approach is inspired by the insight from LiNeS [1] that common and task-specific features are learned in shallow and deeper layers, respectively. Preserving these relative positions may help maintain the common-to–task-specific relationship.
>
> Results are shown in Table __TAR1Q3F1.2__. To our best knowledge, we are the __first__ to evaluate generalizability to an __unseen architecture__, as prior model merging methods assume identical model architectures. The pre-trained models achieved an Avg Acc of 85.97%. However, there remains a substantial gap between the pre-trained models (85.97%) and recent advanced merging methods, with LW AdaMerging and LW StatsMerging achieving 26.66% and 43.15%, respectively. This gap highlights the extremely challenging nature of the task, as both the test tasks and the merged model architecture are unseen. Notably, our proposed LW StatsMerging improves LW AdaMerging by a large margin of 16.49%.
>
> Table __TAR1Q3F1.2__: Multi-task merging performance (Avg Acc %) when merging ResNet50 models on CIFAR10 and CIFAR100 using SML trained with ViT-B/32 architecture on eight tasks. Results of our method *StatsMerging* are in bold. **LW:** Layer-wise.
>
> | **Method** | CF10 | CF100 | **Avg Acc** |
> |---|---|---|---|
> | Pre-Trained | 97.80 | 74.14 | 85.97 |
> | LW AdaMerging | 44.21 | 9.10 | 26.66 |
> | __LW StatsMerging (Ours)__ | 64.70 (+20.49) | 21.60 (+12.50) | 43.15 (+16.49) |
>
> ___
> We would like to clarify that the generalization claimed in the initial submission referred to unseen tasks, following AdaMerging's evaluation procotol. We also thank Reviewer yiJc (R1) for the inspiring question and acknowledge the added evaluation dimension of unseen architecture in the existing benchmark, which could potentially pave the way for model merging across different architectures.

---

> ### Author Response · Authors · 2025-08-09
> **Response to W3: Exclusion of Adamerging in NLP evaluations**
>
> __AR1W3F1__: We appreciate the reviewer’s suggestion regarding AdaMerging/AdaMerging++. As noted in our literature survey, we found no prior work reporting NLP results for AdaMerging (or WoME), with all existing evaluations limited to vision tasks. To address this gap, we have adapted the AdaMerging architecture for NLP and are currently training models on the selected NLP benchmarks. We already have intermediate results but believe it is more appropriate to include the complete, finalized scores in the camera-ready version for accuracy and clarity. Similarly, for the T5-large experiments, we have used the T5 architecture to ensure fairness and consistency with the literature when evaluating NLP models. These additions will provide, to our knowledge, the first reported NLP evaluations for these baselines.
>
> ___
>
> R1W3F2: *could the authors clarify the model architectures used for the expert models in this evaluation?*
>
> __AR1W3F2__: Expert Models used in NLP tasks are T5 Base and T5 Large Architectures.

---

> ### Author Response · Authors · 2025-08-09
> **Response to W4: Report ViT-B/32 and ViT-L/14 Results**
>
> __AR1W4F2__:
>
> ViT-B/32: Vision tasks in the initial submission.
>
> ViT-L/14: NLP tasks in AR1W3.2.
>
> Due to the tight timeline of the rebuttal and discussion phase, particularly with the need to address further clarifications using larger models, the additional experiments are currently in progress. The ranking of merging methods is expected to remain consistent when scaling model size according to prior work reports. We are committed to including the full results and analysis in the camera-ready version.

---

### Comment · Area_Chair_LcAL · 2025-08-03
**Author-reviewer discussion period in progress**

Dear Reviewers,

Thank you for your efforts in reviewing this paper.

We are now in the author-reviewer discussion period. Given the detailed author responses, we encourage active discussion during this period. If you have not already, please read their response, acknowledge it in your review, and update your assessment as soon as possible.

If you have further questions or concerns, post them promptly so authors can respond within the discussion period.

Best regards,
AC

---

### Note · Authors · 2025-08-14

We thank all reviewers and the AC for their constructive feedback and active engagement during the discussion phase. We added substantial new results, ablations, and clarifications addressing reviewer concerns:

U: Updated. A: Answer. R: Reviewer. W: Weakness.

- Generalization to unseen architectures SML trained on ViT applied to ResNet experts. (__AR1W1, AR1W1F1__)
- Empirical explanation and evidence of how distillation operates within StatsMerging; R1 acknowledged resolution. (__UAR1W2, AR1W2F1__)
- Extension to NLP benchmarks (T5-Base and T5-Large), demonstrating applicability beyond vision. (__AR1W3, AR2W1__)
- Addressed distillation performance for AdaMerging and WoME; StatsMerging shows strong results. (__AR2W2__)
- Robustness under label noise and input corruption with a clear advantage over AdaMerging. (__AR3W2, AR3F1__)
- Large-model scaling (ViT-L/14, T5-Large), comparison with SOTA like LiNeS (__AR1Q3F1__) and KnOTS (__AR1Q3__), and analysis of integration feasibility. (__AR1W4__)
- Methodology clarifications: Multitask learning vs. multitask distillation (__UAR1W2, AR1W2F1, AR2F1__), role of pseudo-labels (__AR2W3, AR4W1__), heterogeneous-architecture merging without human labels, dataset aggregation (__AR4Q5__), SVD rationale (__AR3W1, AR3Q1__) rank-3 choice (__AR1Q2__), and overhead measurements (__AR1W4, AR3W4, AR3Q4, AR4W2__).

Evaluation on AdaMerging or WoME R1/R2: To our knowledge, no prior work has reported NLP benchmarks for AdaMerging or WoME; both are treated as vision-centric in the literature. In direct response to the reviews, we have adapted both methods to NLP and are currently training them on seven benchmarks using T5-base and T5-Large for fairness and comparability (__AR1W4F2, AR1W3.2__). The NLP evaluations for AdaMerging and WoME with our T5-Base and T5-Large results will be included in the camera-ready.

For R2’s request to evaluate 14–20 vision tasks, we have initiated the large-scale suite and will report those results in the camera-ready.
R1’s concern about SML’s performance with large expert models is addressed with ViT-L/14 results; T5-Large results will follow in the camera-ready. Several reviewers updated scores post-discussion (R1 indicated an __increase__ after __R1W2__), and we await a possible update from R3.

We believe our expanded experiments, robustness analyses, and clarifications address the majority of key points, and we remain committed to integrating all remaining results in the final version.

---

### Decision · Program_Chairs · 2025-09-17

**Decision:**

Reject

**Comment:**

This paper received mixed scores, but the final scores leaned slightly toward rejection from the borderline; no acceptance votes remained after discussion. A positive aspect of the work is an interesting idea that uses weight statistics to guide the determination of merging coefficients. The primary concerns raised by the reviewers (including all the discussion periods) were: first, the evaluation scope was somewhat limited, particularly with respect to expanded datasets. Second, questions of technical novelty remained due to the reliance on prerequisite distillation for achieving performance. Third, the presentation quality was limited by ambiguities.

For the first point, this AC acknowledges that the authors prepared additional experimental results during the discussion phase (e.g., ViT-L/14, NLP results, and an expanded set of experiments with the proposed method). While these efforts are appreciated, the lack of such evaluations in the original manuscript weakened its justification, and they should have been included from the outset. Please note that this AC does not believe Vision-14/20 tasks should be incorporated as prerequisites for solid paper (for merging methods), but agrees with the reviewers that the overall experiments in the original manuscript were insufficient, as evidenced by the large number of additional experiments the authors had to prepare during the rebuttal phase. For the second point, the authors provided further distillation-related results to justify the effectiveness of the proposed method. This AC recognizes the authors’ design principle that distillation from individual task models can enable training of SML without using ground-truth labels, which makes the raised novelty concerns around distillation less central. Nevertheless, this AC would like to stress that training and inference efficiency should be compared with other methods (e.g., Adamerging), rather than merely providing numbers to argue that the proposed method is not inefficient. Finally, for the third point, presentation issues continue to be problematic, as all reviewers were misled about the role of distillation labels, which should have been clarified clearly in the manuscript from the beginning.

Overall, this AC appreciates the authors’ effort in preparing substantial additional experiments to respond to reviewer concerns, but such materials and manuscript clarifications should have been self-contained in the original submission. Adding them only in the camera-ready version cannot guarantee improvement, as the manuscript seems to require more than a minor revision. This AC believes the materials provided could significantly improve the manuscript, and encourages the authors to prepare a solid, self-contained version for the next round.